# Hcp of the Type VI Secretion System (T6SS) in *Acidovorax citrulli* Group II Strain Aac5 Has a Dual Role as a Core Structural Protein and an Effector Protein in Colonization, Growth Ability, Competition, Biofilm Formation, and Ferric Iron Absorption

**DOI:** 10.3390/ijms23179632

**Published:** 2022-08-25

**Authors:** Nuoya Fei, Weiqin Ji, Linlin Yang, Chunyan Yu, Pei Qiao, Jianpei Yan, Wei Guan, Yuwen Yang, Tingchang Zhao

**Affiliations:** 1Department of Plant Pathology, Plant Protection College, Shenyang Agricultural University, Shenyang 110866, China; 2State Key Laboratory for Biology of Plant Diseases and Insect Pests, Institute of Plant Protection, Chinese Academy of Agricultural Sciences, Beijing 100193, China

**Keywords:** *Acidovorax citrulli*, T6SS, Hcp, activation, multiple functions

## Abstract

A type VI secretion system (T6SS) gene cluster has been reported in *Acidovorax citrulli*. Research on the activation conditions, functions, and the interactions between key elements in *A*. *citrulli* T6SS is lacking. Hcp (Hemolysin co-regulated protein) is both a structural protein and a secretion protein of T6SS, which makes it a special element. The aims of this study were to determine the role of Hcp and its activated conditions to reveal the functions of T6SS. In virulence and colonization assays of *hcp* deletion mutant strain Δ*hcp*, *tssm* (type VI secretion system membrane subunit) deletion mutant strain Δ*tssm* and double mutant Δ*hcp*Δ*tssm*, population growth was affected but not virulence after injection of cotyledons and seed-to-seedling transmission on watermelon. The population growth of Δ*hcp* and Δ*tssm* were lower than *A. citrulli* wild type strain Aac5 of *A. citrulli* group II at early stage but higher at a later stage. Deletion of *hcp* also affected growth ability in different culture media, and the decline stage of Δ*hcp* was delayed in KB medium. Biofilm formation ability of Δ*hcp*, Δ*tssm* and Δ*hcp*Δ*tssm* was lower than Aac5 with competition by prey bacteria but higher in KB and M9-Fe^3+^ medium. Deletion of *hcp* reduced the competition and survival ability of Aac5. Based on the results of Western blotting and qRT-PCR analyses, Hcp is activated by cell density, competition, ferric irons, and the host plant. The expression levels of genes related to bacterial secretion systems, protein export, and several other pathways, were significantly changed in the Δ*hcp* mutant compared to Aac5 when T6SS was activated at high cell density. Based on transcriptome data, we found that a few candidate effectors need further identification. The phenotypes, activated conditions and transcriptome data all supported the conclusion that although there is only one T6SS gene cluster present in the *A*. *citrulli* group II strain Aac5, it related to multiple biological processes, including colonization, growth ability, competition and biofilm formation.

## 1. Introduction

Bacterial fruit blotch (BFB) of cucurbits is a destructive disease that has threatened global cucurbit production for over 50 years since its initial report in 1965 [1,2]. BFB is caused by the Gram-negative bacterium *Acidovorax citrulli* [3,4]. Based on genetic and biochemical traits, *A. citrulli* strains have been divided into two distinct groups (I and II). Group I strains have been mainly isolated from various non-watermelon hosts, while group II strains have been generally isolated from, and are highly virulent, on watermelon (*Citrullus lanatus*) [5,6,7,8]. At present, many virulence factors have been studied in *A*. *citrulli*, such as type III secretion system (T3SS), Type 4 pili (T4P), quorum sensing, toxin-antitoxin systems, the type VI secretion system (T6SS), several two-component regulatory systems, and some other virulence genes [9,10,11,12,13,14,15].

T6SS is one of the bacterial secretion systems that secrete toxic proteins to the outer environment. T6SS has been found in 25% of Gram-negative bacterial species and plays an important role in pathogenicity, biofilm formation, metal ion transport, bacterial interactions, and other biological functions [16,17,18]. The number of T6SS gene clusters in different bacteria can be different. *Pseudomonas aeruginosa* has three T6SS gene clusters which play different functions and secrete different effectors and is studied as the model for T6SS [19]. There are six sets of T6SS gene cluster in *B*. *thailandensis*. A T6SS gene cluster containing 17 genes has been reported in *A*. *citrulli* [20] that is homologous with T6SS-6 in *Burkholderia thailandensis* [21]. The T6SS-1, 2, 4, and 6 of *B. cenocepacia* has antibacterial activity while T6SS-5 affects virulence [22]. This is evidence that T6SS plays multiple function.

The functions of the 17 T6SS genes have been studied individually in *A*. *citrulli* group I strain xjl12 [20]. All 17 mutants showed no significant difference in virulence to its natural host melon, and only four single gene deletions Δ*vasD*, Δ*impK*, Δ*impJ*, and Δ*impF* reduced melon seed to seedling transmission and biofilm formation [20]. Furthermore, Levy et al. found that a new T6SS effector family, Hyde proteins, mediated antibacterial activities of *A. citrulli* group II strain AAC00-1. However, the deletion of five *Hyde1* loci in AAC00-1 did not affect the virulence on host watermelon [23]. Recently, Pei et al. [24] identified the T6SS antibacterial effector RhsB in *A. citrulli* strain AAC00-1, which requires the upstream-encoded chaperones EagT2 and VgrG3 to accomplish its secretion. However, whether one T6SS gene cluster of *A*. *citrulli* can play all the functions that reported in other bacteria, such as pathogenicity, biofilm formation, metal ion transport, bacterial interactions, and other biological functions, still needs further study.

Hcp (Hemolysin co-regulated protein, also known as TssD) is both a special effector and a structural member of the inner tube of the T6SS [25]. Because of the secretory function and the assembly of the secretory channel with VgrG (Valine-glycine repeat protein G; TssI) and PAAR (Pro-Ala-Ala-Arg-repeat-containing protein) [26,27,28], Hcp and VgrG can be used to study T6SS activation [29,30,31,32,33]. In a preliminary study, we found that the expression of one of the T6SS *vgrG* genes *Aave_2840* was upregulated when the T3SS gene *hrpE* was absent in the *A*. *citrulli* group II strain Aac5 (unpublished transcriptome data). When the Type 4 pilus (T4P) gene *pilA* was deleted, the expression of both *hcp* and *vgrG* was affected (unpublished transcriptome data). These unexpected data concerning the T6SS gene indicate that T6SS of *A*. *citrulli* may have multiple biological functions, which needs to be proven. Additionally, since TssM (type VI secretion system membrane subunit, also known as VasK) is a key component of the T6SS apparatus [16] and present in the middle of the A. citruli T6SS gene cluster [20], we used a *tssm* mutant as a T6SS-deficient control.

At present, there is no report on the conditions that activate the T6SS in *A*. *citrulli*, which is in need for further research such as effector identification and transcription regulation of T6SS. In addition, due to the substantial differences between the two groups of *A*. *citrulli* strains and the horizontal transfer of T6SS genes [34,35], the function of the T6SS in *A*. *citrulli* may vary by groups and strains. T6SS may also be related to other biochemical processed to realize its multiple functions.

T6SS may also related to other biochemical processed to realize its multiple functions. In a preliminary study, we found that the expression of one of the T6SS *vgrG* genes *Aave_2840* was upregulated when T3SS gene *h**rpE* was absent in the *A*. *citrulli* group II strain Aac5 (unpublished transcriptome data). When T4P gene *pilA* was deleted, the expression of both *hcp* and *vgrG* was affected (unpublished transcriptome data). These unexpected data of T6SS gene indicate that T6SS of *A*. *citrulli* may have multiple biological functions, which needs to be proven.

In this study, we investigated the role of the key element *hcp* in virulence, competition, biofilm formation, growth ability and ferric ion absorption to better understand T6SS in *A*. *citrulli*. Because there are 12 VgrGs but only one Hcp in *A*. *citrulli*, we selecedt Hcp as a marker for studying T6SS activation and secretion in this study. The transcriptional orientation of the *hcp* gene differs from that of all the other genes in the T6SS cluster of *A*. *citrulli*, which makes the *hcp* gene unique. Additionally, since TssM is a key component of the T6SS apparatus [16] and present in the middle of the *A*. *citruli* T6SS gene cluster [20], we used a *tssm* mutant as a T6SS deficient control because the phenotypes of Δ*tssm* has already been reported in both Tian et al. (phenotypes of pathogenicity) and Pei et al. (phenotypes of competition) [20,24]. The activation of *hcp* at different conditions was determined by qPCR and Western blotting to reveal the conditions that activate the T6SS in *A*. *citrulli* strain Aac5. In addition, the transcriptome data under the T6SS-activated condition was analyzed to study the possible relation between T6SS and other systems and to find candidate effectors. These results will support further research on potential T6SS effectors and the interaction between Hcp and VgrG proteins in *A*. *citrulli*.

## 2. Results

### 2.1. Sequence Analysis of the T6SS and Hcp Homolog in A. citrulli

Based on the genome data on NCBI, we chose the whole T6SS gene clusters predicted in the genome of four *A. citrulli* representative strains (group II strains AAC00-1 which were isolated from *Citrullus lanatus* var. *citroides*, KACC17005 isolated from *Citrullus lanatus*; group I strains M6 and pslb65 isolated from *Cucumis melo*) to conducted sequence alignment. We found that the T6SS sequences of each strain has its own single nucleotide polymorphism (SNP) which can lead to amino acid mutations (Appendix A). This indicated that although T6SS is conserved, T6SS of different strains can be different.

The Hcp sequence from *A. citrulli* group II Aac5 was 100% identical to *Aave_1465* from *A. citrulli* group II AAC00-1 and *APS58_3209* from *A. citrulli* group I M6. (Figure 1a). The *Acidovorax avenae* Hcp sequence is 95% identical to Hcp from *A. citrulli*. The phylogenetic tree showed that Hcp is highly conserved within *A. citrulli* (Figure 1b). While *A. citrulli* strains have one copy of Hcp, certain species have more than two copies of Hcp. Interestingly, six Hcp from *B. thailandensis* clustered in different clades [21] (Figure 1b). All Hcp of *A. citrulli* and Hcp of T6SS-6 of *B*. *thailandensis* were clustered in the same clade, which is consistent with a previous study [21]. This result demonstrates the consistent evolution of Hcp and T6SS gene cluster.

### 2.2. Construction of Δhcp, Δhcpcomp, Δtssm, and ΔhcpΔtssm

PCR amplification using the primer pair *hcp*-TF/*hcp*-TR yielded a 352-bp amplicon when using DNA from WT Aac5, but no amplicon when using DNA from the *hcp* deletion mutant Δ*hcp* (Appendix A). PCR amplification using DNA from WT strain Aac5 and Δ*hcp* yielded 1271-bp and 752-bp PCR amplicons, respectively, with the primer pair *hcp*-1F/*hcp*-2R (Appendix A). The primer for *tssm* deletion mutant was designed based on *Aave_1473* of AAC00-1 and was confirmed using the same approach. A 1156-bp fragment of the entire *hcp* open reading frame (ORF) with its native promoter was amplified and fused with pBBRNolac-HA to obtain the complemented strain Δ*hcp*comp. The strains Δ*hcp*, Δ*tssm*, Δ*hcp*Δ*tssm*, and Δ*hcp* complemented strain Δ*hcp*comp (Appendix A) were confirmed by sequencing and the strains with HA tag Δ*tssm*-*hcp*-HA, Aac5-hcp-HA, and Δ*hcp*-*hcp*-HA-pBBR were confirmed by western blotting.

### 2.3. T6SS Affects the In-Vitro Growth of A. citrulli and Its Ability to Absorb Ferric Iron

At early and middle stage (lag and logarithmic phase), the growth curve of Δ*hcp* was basically consistent with WT strain Aac5 in KB medium. At the late stage of growth (stationary phase) in KB medium, Δ*hcp* showed no sign of decline. The cell density by Optical Density value when measured at 600 nm (OD_600_) of this strain was significantly higher than the other strains (*p* = 0.05) at the late stage of growth (stationary phase), which indicated that *hcp* may be related to cell density regulation (Figure 2a,b). In M9 medium, however, Δ*hcp* grew slower than WT Aac5 (*p* = 0.05) (Figure 2c). The growth abilities of Δ*tssm* and Δ*hcp*Δ*tssm*, both in KB, were significantly lower than the other strains (Figure 2a,b). Compared to the WT strain, Δ*hcp* was significantly less effective in absorbing ferric iron (*p* = 0.05) (Figure 2d).

### 2.4. Δhcp, Δtssm, and ΔhcpΔtssm Strains Showed Differences in Colonization Ability but Not in Inducing Symptoms

Population growth and virulence on cotyledons. To assess the effect of *hcp* on the virulence of *A. citrulli* strain Aac5, watermelon cotyledons were injected with cell suspensions of *A*. *citrulli* strain Aac5, Δ*hcp*, Δ*hcp*comp, Δ*tssm*, and Δ*hcp*Δt*ssm*, respectively. When the initial inoculum concentration was 10^4^ CFU·mL^−1^, there were no visible symptoms until 120 h post-injection. In the first 24 h, Δ*tssm*, Δ*hcp*Δ*tssm*, Δ*hcp*, and the Hcp-complemented strain Δ*hcp*comp could not colonize the host tissue at the injected cell concentration, while only the WT strain Aac5 maintained the initial cell concentration (Figure 2b). All strains then began to grow after the first 24 h, but the bacterial populations of Δ*hcp* and Δ*hcp*Δ*tssm* were significantly lower than the other strains at 48 h post-injection (*p* = 0.05). After 72 h, the bacterial populations of Δ*hcp* were significantly higher than that of Aac5, while the bacterial populations were not significantly different between Δ*tssm* and Aac5 until 96 h (*p* = 0.05). The bacterial populations of Δ*hcp*Δ*tssm* remained lower than that of the other strains from 48 h to 120 h (Figure 3a,b). When the initial inoculum concentration was 10^6^ cfu/mL, the symptoms were evident at 72 h post-injection. All strains maintained the initial concentration of 10^6^ in the first 24 h. After 48 h, the bacterial populations of Δ*tssm* was significantly higher than Aac5, while the bacterial population of Δ*hcp* was significantly higher than Aac5 until 72 h (*p* = 0.05). However, this was not the case for Δ*hcp*Δ*tssm*, and the symptoms caused by Δ*hcp*Δ*tssm* were less severe than those induced by the other strains after 120 h (Figure 3c,d). These results suggest that Δ*hcp*, Δ*tssm*, and Δ*hcp*Δ*tssm* showed differences in their ability to colonize watermelon cotyledons. When *hcp* or *tssm* were deleted separately, the decline phase was delayed.

When the concentration of bacterial suspension was 10^4^ CFU·mL^−1^, the cotyledons showed only mild symptoms (disease severity was about 1 to 3) [36,37] and the bacterial population was in the growth stage. When the concentration was 10^6^ CFU·mL^−1^, the bacterial population started to decrease after 72 h. In our virulence assays, deletion of *hcp* and *tssm* did not significantly affect the disease severity on cotyledons when the initial concentrations were 10^4^ CFU·mL^−1^ and 10^6^ CFU·mL^−1^ (data not shown, *p* = 0.05). When the OD_600_ was 0.3, the concentration of *A. citrulli* was approximately 3 × 10^8^ CFU·mL^−1^ and log CFU·mL^−1^ was approximately 8.48. When the OD_600_ value was 1.5, the concentration was approximately 2 × 10^9^ CFU·mL^−1^ and log CFU·mL^−1^ was approximately 9.30. As the result in Figure 2b,d show, the bacterial population in watermelon cotyledons could not reach to 2 × 10^9^ CFU·mL^−1^, the same population as OD_600_ = 1.5 in KB medium, until the cotyledons showed complete necrosis.

Population growth after seed-to-seedling transmission. Compared with ddH_2_O, the delay of seed germination was easily observed when inoculated with *A. citrulli* strains. The seedling samples could not be collected until 12 d. Watermelon seedlings inoculated by all strains showed significant effects on weight and root lengths, but there was no significant difference among wild type Aac5, Δ*hcp*, Δ*hcp*comp, Δ*tssm*, and Δ*hcp*Δ*tssm* (data not shown). At the early stage (12 d and 15 d) of germination, the bacterial populations for all mutant strains were significantly lower than wild type (*p* = 0.01). However, the bacterial populations of Δ*hcp*, Δ*tssm*, and Δ*hcp*Δ*tssm* at 18 d were all higher than wild type (*p* = 0.01) (Figure 3e,f). The seedlings treated by wild type Aac5 stopped growing after 18 d. The growth of bacterial population in seedlings was consistent with the results in watermelon cotyledons.

### 2.5. T6SS Affected the Biofilm Formation Ability

When grown in KB medium, Δ*hcp*, Δ*tssm*, and Δ*hcp*Δt*ssm* formed more biofilm than WT strain Aac5 (*p* = 0.05) (Figure 4a,e). The results were similar in M9 when starting from OD_600_ = 0.3. However, Δ*tssm* and Δ*hcp*Δt*ssm* but not Δ*hcp* formed more biofilm than Aac5 in M9 when starting from OD_600_ = 1.5 (*p* = 0.05) (Figure 4b,e). Biofilm of all strains increased significantly in a ferric iron environment (*p* = 0.05) (Figure 4c,e). When co-cultured with DH5α, all strains formed more biofilm compared to the without DH5α treatment (*p* = 0.05). The WT and Δ*hcp*comp groups formed significantly more biofilm than Δ*hcp*, Δ*tssm*, and Δ*hcp*Δt*ssm* (*p* = 0.05) (Figure 4d,e). Besides, Δ*hcp*comp was consistent with WT strain Aac5 in KB medium when the OD value was 1.5 but not 0.3. The condition of OD_600_ = 1.5 in KB medium proved to be the activated condition of Hcp in later experiments. Biofilm formation ability of Aac5 in the activated condition of cell density and competition was significantly higher than in the inactivated condition.

### 2.6. Acidovorax citrulli T6SS Is Involved in Interspecies and Intraspecies Competition

After co-culture with *A. citrulli* strains (*Ac* strains for short) Aac5, Δ*hcp*, Δ*hcp*comp, Δ*tssm*, and Δ*hcp*Δ*tssm*, the colony counts of *E. coli* DH5α and *P. aeruginosa* ATCC27853 were significantly decreased compared to the ddH_2_O control (*p* = 0.05) (Figure 5a,c). In experiments with the weakly competitive strain *E. coli* DH5α, the anti-bacterial activity of *A. citrulli* was immediately apparent after co-culture (Figure 5a). However, the anti-bacterial activity became evident at 4 h post co-culturing with the strongly competitive strain *P. aeruginosa* ATCC27853 (Figure 5c). The anti-bacterial activities of the Δ*hcp*, Δ*tssm*, and Δ*hcp*Δ*tssm* deletion mutants were significantly lower than WT Aac5 (*p* = 0.05). The strain’s survival was not significant when co-cultured with *E. coli* DH5α (Figure 5b). When the prey was *P. aeruginosa* ATCC27853, the bacterial populations for Δ*hcp*, Δ*tssm*, and Δ*hcp*Δ*tssm* were significantly lower than Aac5 and the *hcp*-complemented strain Δ*hcp*comp (*p* = 0.05) (Figure 5d). The anti-fungal activity of *A. citrulli* strains was evident at 12 h after culturing with both *S. cerevisiae* Y2HGold and *P. pastoris* GS115, and the survival of the *A. citrulli* strains was barely affected (Figure 5e–h).

For intraspecies competition, *A. citrulli* group I strain pslb65-pBBRMCS5, which shares a similar T6SS with Aac5, was used as the prey. The differences in pslb65-pBBRMCS5 survival after co-culture with the group II strains Aac5, Δ*hcp*, Δ*hcp*comp, Δ*tssm*, and Δ*hcp*Δ*tssm* were not significant. The survival of pslb65-pBBRMCS5 showed no significant difference in ddH_2_O CK of 0 h and 3 h (Figure 6a). Compared to the log_10_CFU·mL^−1^ of 3 h, all survival of pslb65-pBBRMCS5 after co-culture with strains except Δ*hcp*Δ*tssm* showed significant reduction. Besides, the survival of the Δ*hcp* and Δ*hcp*Δ*tssm* strains was significantly lower than the other Aac5 derivatives (*p* = 0.05) (Figure 6b).

The T6SS of *A. citrulli* affected the anti-fungal and anti-bacterial activities as well as their own survival in competition assays. When *hcp* was deleted, the strain generally showed significantly lower activity in the anti-bacterial competition assay but not in the antifungal assay. The strains Δ*hcp*, Δ*hcp*comp, Δ*tssm*, and Δ*hcp*Δ*tssm* showed similar phenotypes in the interspecies competition assays but not in the intraspecies competition assays. The survival of Δ*hcp*Δ*tssm* and Δ*hcp* were both significantly less than the survival of Aac5, while the reduced survival of Δ*tssm* was not statistically significant.

### 2.7. Hcp Can Be Activated by Cell Density, Ferric Iron, Competition, and the Host

Activated conditions of Hcp. Based on our pre-test, Hcp of *A*. *citrulli* group II strain Aac5 was not activated when cultured in KB medium until its OD_600_ value reached 0.8 (Appendix A), which means T6SS was also not activated in the meantime. To explore Hcp activation conditions, the following factors were examined: cell density, competition and host condition. The expression of *hcp* was significantly higher at a cell density of OD_600_ = 1.5 than OD_600_ = 0.3 and 1.0 when cultured in KB (Figure 7b). When colonizing the host tissues, the expression of *hcp* was also significantly higher at 48 and 72 h post inoculation (hpi) than at 24 hpi (*p* value < 0.0001) (Figure 7c). The initial concentration in planta was 10^6^ CFU·mL^−1^ (OD_600_ = 0.3 equates to 3 × 10^8^ CFU·mL^−1^ and OD_600_ = 1.5 equates to 2 × 10^9^ CFU·mL^−1^), when *hcp* activated at 72 h in planta, the concentration approached 3 × 10^8^ CFU·mL^−1^ but was still significantly lower than 2 × 10^9^ CFU·mL^−1^. When co-cultured with *E. coli* DH5α, the expression of *hcp* in Aac5 showed significant difference compared with the non-activation conditions (Figure 7d).

We designed the activated condition assay based on the expression difference between wild type and complementary strain Δ*hcp*comp (Figure 7a and Appendix A). Thus, we used Δ*hcp*comp as a positive control of activation in qPCR and Western blot assay. When *hcp* was activated in wild type strain by cell density and host condition, the expression could still not reach the level of Δ*hcp*comp (Appendix A). Only when *hcp* was activated by competition, the expression of *hcp* in Aac5 was significantly higher than in Δ*hcp*comp (Appendix A). Under the activated condition, the different expression of *hcp* between the wild type and Δ*hcp*comp was less than that under the inactivated conditions.

Production and secretion of Hcp. Production and secretion of Hcp in cell lysates and culture supernatants of Δ*tss**m-hcp*-HA, *hcp*-HA-Aac5, and Δ*hcp*-*hcp*-HA-pBBR at two cell densities (OD_600_ = 0.3 and 1.5) were detected by Western blotting. The Hcp protein in *hcp*-HA-Aac5 was fused with an HA tag at the C-terminal end. Δ*hcp*-*hcp*-HA-pBBR was used as the positive control because the expression of Hcp in this strain was constant. The production and secretion of Hcp at OD_600_ = 1.5 was significantly higher than at OD_600_ = 0.3 (Figure 8a,b). The secretion of Hcp also increased in the presence of ferric iron compared to non ferric iron, and in competition assays compared to non-competition assays. (Figure 8b,c).

### 2.8. Transcriptome Analysis under T6SS Activating Conditions Indicated That the T6SS Is Related to Multiple Pathways

The genes that showed differential expression between the WT strain Aac5 and the *hcp* deletion mutant Δ*hcp* at high cell density (OD_600_ = 1.5) were determined by transcriptome sequencing and analysis. The key T6SS gene *vgrG* in Δ*hcp* was down-regulated (Figure 9d). Significant changes were detected in the expression of genes related to bacterial secretion systems, protein export, and a few other pathways in Δ*hcp* compared with Aac5 under T6SS activating conditions (Figure 9a, Appendix A). SecY and SecE of Sec dependent pathway were down-regulated in Δ*hcp* while other genes of Sec dependent pathway were not affected. Secretin, IMP, ATPase and ATPase associated proteins of T3SS were up-regulated in Δ*hcp*. GspH and GspI of T2SS were down-regulated in Δ*hcp* while GspF was up-regulated (Figure 9).

There were twelve genes encoding secreted proteins predicted by SecretomeP-2.0 (https://services.healthtech.dtu.dk/service.php?SecretomeP-2.0, accessed on 27 April 2022) [38] with SecP score above 0.5. Among these twelve genes, *Aave_4148* encode a hypothetical protein with no significant match domain in Pfam (http://pfam.xfam.org/, accessed on 27 April 2022) [39], which can be a candidate of new T6SS effector protein. There were five genes encoding secreted proteins that showed partial homology to effectors proteins recorded in SecReT6 (https://bioinfo-mml.sjtu.edu.cn/SecReT6/blast.php, accessed on 13 May 2022) [40]. There were four genes showed partial homology to component proteins, regulators and accessory proteins recorded in SecReT6. It should be noted that the prediction results by SecretomeP-2.0 and SecReT6 were not always consistent (Appendix A). These results mean that hcp affects the expression and secretion of T6SS effectors.

## 3. Discussion

*Acidovorax citrulli* harbors only one T6SS gene cluster, in which the *hcp* gene (*Aave_1465*) is transcribed in opposite direction. Hcp was reported in *P*. *aeruginosa* as a baseplate complex (BC) structural member, a secretory protein, a chaperone and receptor of substrates that forms the tip of T6SS and binds to effector molecules [25], which makes it a unique component of the T6SS and can be used to study the function and activation of T6SS. TssM is one of the inner membrane proteins of T6SS membrane complex (MC). Deletion of TssM, the gene responsible for membrane complex of T6SS, was used as a T6SS-defecient control in our experiments since the phenotypes of Δ*tssm* has already been reported in both Tian et al. [20] (phenotypes of pathogenicity) and Pei et al. [24] (phenotypes of competition).

By T6SS gene cluster sequence alignment, we found that the T6SS cluster varies among *A. citrulli* strains. Phylogenetic analysis based on Hcp was consistent with the whole T6SS gene cluster [21], which means Hcp in Aac5 may have a similar function as other Hcp proteins of the same clade in the phylogenetic tree. Based on these results, we conducted experiments to study the function of Hcp in Aac5.

Based on the results of virulence, growth, competition, biofilm formation, and ferric ion absorption, we deduced that the phenotype of the double mutant strain Δ*hcp*Δ*tssm* may be more representative of the phenotype of Aac5 when T6SS is disabled. The colonization, growth, competition, biofilm formation, and ferric ion absorption abilities in Δ*hcp*Δ*tssm* were affected, which means T6SS was related to these phenotypes.

The phenotypes of Δ*hcp* and Δ*tssm* were basically consistent in most activated conditions (cell density of OD_600_ = 1.5 in KB medium and host condition after 72 h) we demonstrated (only except the phenotypes in ferric ion). However, they still showed differences in some phenotypes. The BC of T6SS which *hcp* belongs to assembles independently of the MC which *tssm* belongs to [41,42]. The transcriptional orientation of the *hcp* gene differs from that of all other genes in the T6SS cluster of *A. citrulli*. In our secretion assay and Pei’s article [24], the secretion of Hcp was reduced but not fully lost in Δ*tssm* (Figure 8), which might also indicate that Hcp was an independent component in T6SS and could still work outside the cell in Δ*tssm*. In Tian’s article, different T6SS structural gene deletion mutants showed differences in phenotypes. The Δ*vasD*, Δ*impK*, Δ*impJ* and Δ*impF* affected the seed-to-seedling transmission of melon, while other 13 mutant strains of T6SS core genes did not [20]. The T6SS assembly and the mechanism of action are conserved across species, but the repertoire of secreted toxic effectors and cognate immunities and their regulatory mechanisms vary by strains [41,42]. These could explain the differences of *hcp* and *tssm* phenotypes.

The function of T6SS regarding ferric ion absorption and growth abilities has not been reported in *A*. *citrulli*. The function of *hcp* in *A. citrulli* group II strain Aac5 has certain differences from the *hcp* in *A. citrulli* group I strain xjl12 reported by Tian et al. [20]. They reported that the *hcp* in xjl12 did not affect colonization in melon and biofilm formation. In our study, Δ*hcp* showed significant differences in colonization in watermelon, growth and biofilm formation abilities compared to WT.

In the competition assay, Δ*hcp* and Δ*hcp*Δ*tssm* showed significant differences in anti-bacterial activity but not in anti-fungal activity, which is different from the conclusion reached by Pei et al. [24]. The T6SS in *A. citrulli* group II strain AAC00-1 showed both anti-bacterial activity and anti-fungal activity, and Δ*tssm*, Δ*rhsB* and Δ*rhsE* showed lower anti-bacterial activity and anti-fungal activity [24]. These results support our hypothesis that T6SS functions vary among strains. The Δ*hcp* showed no sign of decline when the WT strain and complementary strain started to decline from 70 h to 84 h in KB at the initial concentration of 10^7^ CFU·mL^−1^, which indicates that Δ*hcp* may lose its ability to control cell density. Therefore, the cell density of Δ*hcp* was higher than that of WT strain Aac5 both in the stationary stage of growth in culture and in the late colonization stage (96 hpi post injection when initial concentration is 10^4^ CFU·mL^−1^, 72 hpi when initial concentration is 10^6^ CFU·mL^−1^ in cotyledons and 18 d when initial concentration is 10^6^ CFU·mL^−1^ in seedlings) in host tissues. Δ*hcp*, Δ*tssm*, and Δ*hcp*Δ*tssm* showed reduced competition abilities. *hcp* can be secreted and the phenotype except the growth and colonization ability of Δ*hcp* was basically consistent with Δ*hcp*Δ*tssm*.

Our previous studies showed that when the WT strain Aac5 was cultured to OD_600_ = 0.8 in KB medium, the expression of *hcp* was significantly lower than in the stably expressed complemented strain Δ*hcp*comp (Appendix A). Therefore, Hcp and the T6SS might not be activated under this growth condition. It is critical to define the conditions necessary for T6SS activation for further research, such as screening, identifying, and characterizing T6SS effectors. In addition, the activation conditions could help to confirm the function of T6SS. In this study, we determined that a cell density of OD_600_ = 1.5 could activate the T6SS. The expression and secretion of Hcp were significantly higher in cultures grown to OD_600_ = 1.5 than in cultures under inactivated conditions. Moreover, in the late colonization stage in the host at 72 hpi, the expression of Hcp was also higher than that in the early stage (24 hpi). The cell density at 72 hpi in the host did not reach the same level as that of OD_600_ = 1.5 in KB medium (Figure 2d, when the OD_600_ value was 1.5; the concentration was approximately 2 × 10^9^ CFU·mL^−1^ and log CFU·mL^−1^ was approximately 9.301029996, so the cell density at 72 hpi in the host did not reach the same level as that of OD_600_ = 1.5), which means that Hcp and T6SS might also be activated by host condition. However, assays on secretion of Hcp in host remain to be explored and established. Hcp was also activated and expressed more when *A*. *citrulli* was co-cultured with *E. coli* DH5α, which led to the same conclusion as in our phenotype experiment that T6SS affected competition ability. In conclusion, T6SS can be activated by cell density, ferric ions, competition and host condition, and might affect various biological processes such as growth, competition, biofilm formation, and ferric ion absorption. The *Burkholderia mallei* and *B*. *pseudomallei* cluster 1 T6SS are negatively regulated by iron and zinc [43], but in Aac5 we found that the production and secretion of Hcp was promoted by ferric irons (Figure 8c). In the plant pathogen *P*. *syringae* pv. tomato DC3000, the expression of *hcp*2 is not induced in planta and not related to virulence or colonization, but required for survival in competition [44].

Based on qPCR and Western blotting, we demonstrated that the expression of *hcp* in Δ*hcp*comp is stable. The survival and competition ability of Δ*hcp*comp is consistent with WT strain Aac5, whereas other phenotype such as biofilm formation, colonization, ferric iron absorption and growth ability of Δ*hcp*comp partial recovery, which may be due to: the following. (1) In some conditions T6SS was not activated. This was supported in that the phenotypes of wild type strain and Δ*hcp*comp basically were consistent in most activated conditions (cell density of OD_600_ = 1.5 and host condition after 72 h) we demonstrated (only except the phenotypes in ferric ion). (2) Hcp in complementary strain was in an expression vector which could express without activation, even when not in the genome. Although the expression plasmid had a low copy number, as our pre-test, when *hcp* was activated in the wild type strain the expression could still not reach the level of Δ*hcp*comp (except when co-cultured with DH5α, and only under this condition was expression of *hcp* in Aac5 significantly higher than that in Δ*hcp*comp). However, the differential expression of *hcp* between wild type and Δ*hcp*comp was reduced significantly in the activated condition (Appendix A). The difference in expression of *hcp* should be responsible for the phenotypes.

We analyzed the genes that showed differential expression between Aac5 and Δ*hcp* at OD_600_ = 1.5 by transcriptome analysis. Some T6SS genes, such as the VgrG coding gene *Aave_2840* were differentially expressed in the T3SS gene deletion mutant of *A. citrulli* Aac5 (unpublished transcriptome data). Data in this study also support this result in that T3SS and T4P may have some association with T6SS (Figure 9d and Appendix A), but this association has yet to be studied in depth. In transcriptome data based on T6SS activated conditions, only 29 genes showed significantly different expression by both log_2_FoldChange and *p* value in Δ*hcp* from Aac5.

Among the differentially expressed genes under the cell density activated condition, we found twelve genes encoding secreted proteins predicted by SecretomeP-2.0. Among them *Aave_4148* encodes a hypothetical protein with no significant known domain, which could be a candidate of new type of T6SS effector. However, further experiments such as secretion assay and function analyses need to be conducted to identify it as a T6SS effector. There were five genes encoding secreted proteins that showed certain homology to effectors proteins recorded in SecReT6. The homogeneity of some genes was not high enough, and also need further experiments of identification. These data support the multiple functions of T6SS and might help to identify T6SS effector proteins with certain functions under defined growth conditions. However, we have to consider that *A. citrulli* T6SS has antibacterial and antifungal activities. If the antibacterial and antifungal activities of effectors were too strong, some experiments to identify effectors could be difficult to conduct, such as yeast two hybrid and bacterial two hybrid assays, as well as prokaryotic and eukaryotic expression.

Previous research on bacterial type VI secretion systems showed that VgrG can participate in the assembly at the tip of the T6SS with Hcp and PAAR [29]. According to Tian et al. (2015) [1], there is only one *hcp* gene that is part of the only T6SS gene cluster present in *A*. *citrulli*. However, there are twelve *vgrG* genes which are not in the T6SS gene cluster. This phenomenon might be related to the multiple functions of T6SS. Unlike *P*. *aeruginosa* [18], *A*. *citrulli* might have multiple functions for one T6SS by employing different types of VgrG proteins rather than through multiple sets of T6SS. Recently, Pei et al. [22] identified a T6SS antibacterial effector RhsB that requires the upstream-encoded chaperones EagT2 and VgrG3 for its secretion. Further study of the 12 *vgrG* genes based on the interaction between Hcp and VgrG could help identify functional VgrG proteins. Therefore, it is important to determine the function of Hcp and to define its key function in the T6SS. The T6SS activation conditions can also be used for the identification and functional research of VgrG.

## 4. Materials and Methods

### 4.1. Bacterial Strains, Plasmids, Media, and Culture Conditions

Bacterial and fungal strains used in the study are listed in Appendix A. The *A*. *citrulli* wild-type (WT) strain Aac5 and mutants were routinely cultured at 28 °C on King’s B (KB) medium, *E. coli* DH5α was routinely cultured at 37 °C on Luria-Bertani (LB) medium, and *P. aeruginosa* ATCC27853 was routinely cultured at 28 °C on LB medium. *Saccharomyces cerevisiae* Y2HGold and *Pichia pastoris* GS115 strains were routinely cultured at 30 °C on Yeast Extract Peptone Dextrose Medium (YPDA). Antibiotics were used at the following concentrations: kanamycin (Km, 100 μg/mL), chloramphenicol (25 μg/mL), ampicillin (Amp, 50 μg/mL), gentamicin (Gm, 25 μg/mL), and Aureobasidin A (AbA, 100 μg/mL). The antibiotics were only used in the competition assay of counting colonies in plates and bacteria culture before experiments, but not in the experiments such as growth ability, biofilm formation, virulence and colonization. The media used in competition assays to count the colony forming units per milliliter (CFU·mL^−1^) of *A*. *citrulli* WT strain Aac5 and mutants were KB supplemented with Amp and Km (for competition with other bacteria) and KB supplemented with Amp, Km, and AbA (for competition with Y2HGold and GS115). The medium used to count the CFU·mL^−1^ of prey was KB supplemented with Amp and Gm for pslb65, LB supplemented with Km for DH5α, CFC selective agar for *P. aeruginosa* ATCC27853, and YPDA supplemented with Gm for Y2HGold and GS115.

### 4.2. Bioinformatics Analysis

To verify whether *hcp* homologous genes are present in strain Aac5, we amplified and sequenced *hcp* from Aac5. The predicted amino acid sequences of Hcp from *A. citrulli* strain Aac5, AAC00-1, M6, and *A. avanae* strain ATCC19860 were aligned using ClustalW2 (https://www.ebi.ac.uk/Tools/msa/clustalw2/, accessed on 28 April 2022) and Jalview [45]. The amino acid sequence alignment of representative Hcp proteins was used to construct a phylogenetic tree in MEGA11 using the Maximum Likelihood method and the JTT matrix-based model. Initial tree(s) for the heuristic search were obtained automatically by applying the Neighbor-Joining and BioNJ algorithms to a matrix of pairwise distances estimated using the JTT model and then selecting the topology with superior log-likelihood value. The phylogenetic tree was rooted with the outgroup Hcp2 of RIMD2210633 BAC62370.1, which is relatively low homology with other Hcps.

### 4.3. Construction of hcp and tssm Marker-Less Deletion Mutants, hcp Complementation

All plasmids used in this study are listed in Appendix A. Molecular manipulations were performed using standard procedures [46]. Constructs were ligated using the ClonExpress II One Step Cloning Kit (Vazyme Biotech, Nanjing, China). Primers for polymerase chain reaction (PCR) used in this study (Appendix A) were synthesized by BGI Laboratories (BGI, Shenzhen, China). KOD-Plus neo (Toyobo, Shanghai, China) and 2× Taq Plus PCR MasterMix (Tiangen, Beijing, China) were used for PCR amplification.

To generate the *hcp* marker-less mutant, a 390 bp fragment upstream of the *hcp* open reading frame (ORF) and a 417 bp fragment downstream of the *hcp* ORF were amplified by PCR and introduced into the pK18mob*sacB* vector. To generate the *tssm* marker-less mutant, a 422 bp fragment upstream of the *tssm* open reading frame (ORF) and a 372 bp fragment downstream of the *tssm* ORF were amplified by PCR and introduced into the pK18mob*sacB* vector. The recombinant vectors were transferred into *A*. *citrulli* strain Aac5 by tri-parental mating with an *E*. *coli* strain carrying a helper plasmid, pRK600. Marker-less *hcp* and *tssm* mutants of *A*. *citrulli* strain Aac5 were generated by allele exchange and screened as described by Zhang et al. [47]. The *hcp tssm* double mutant was constructed based on the *hcp* marker-less mutant by deleting *tssm*.

To construct a complementation vector, the complete *hcp* gene (1156 bp), including its native promoter, was amplified with primers *hcp*HBF/*hcp*HBR (Appendix A) and inserted into the shuttle vector pBBRNolac-HA to create the complementation vector pBBR-*hcp*-HA, which was then introduced into the *hcp* mutant to generate the complemented strain Δ*hcp*comp. The strain Aac5-*hcp*-HA was generated based on Aac5 by fusing an HA tag to the C-terminal end of Hcp. The strains Δ*tssm*-*hcp*-HA was generated based on Δ*tssm* by fusing an HA tag to the C-terminal end of Hcp. The expression of *hcp* in both Aac5-*hcp*-HA and Δ*hcp*-pBBR-*hcp*-HA was verified in the complemented strain by Western blotting [46]. The primer pairs *hcp*A/*hcp*B and *hcp*C/*hcp*D were used to amplify and clone a 482-bp fragment upstream of the *hcp* stop codon with an HA tag and a 517-bp fragment downstream of *hcp* with an HA tag, respectively, to fuse an HA tag at the C-terminal of Hcp.

In the competition assay, the plasmid pBBRMSC2 was transferred into all the mutants and wild type Aac5 by tri-parental mating to introduce Km resistance. The plasmid pBBRMSC5 was transferred into DH5α and the *A*. *citrulli* group I strain pslb65 to introduce Gm resistance. All strains were confirmed by PCR and DNA sequencing.

### 4.4. Virulence and Seed-to-Seedling Transmission

The in planta colonization ability was determined in the cotyledons of 2-week-old watermelon seedlings using the method in Guan et al. (2020) [48]. Both high (1 × 10^6^ CFU mL^−1^) and low (1 × 10^4^ CFU mL^−1^) density cultures were used. Thirty cotyledons were inoculated by injection with either *A*. *citrulli* strains or water and collected at 24, 48, 72, 96, and 120 h post inoculation (hpi). The cotyledons were weighed and homogenized in sterile water, and the colony forming units (CFU) per gram were assayed by ten-fold serial dilution plating.

The ability of *A*. *citrulli* strains to colonize watermelon seedlings during germination was determined by seed inoculation. The strains were cultured to OD_600_ = 0.3 (3 × 10^8^ CFU·mL^−1^), washed, and resuspended in sterile water. Eighteen seeds soaked in bacterial suspension for 4 h were transferred to 15 mL centrifuge tubes containing sterile absorbent cotton, filter paper, and sterile water. When the seeds started to germinate, the seedlings were collected and weighed, and the *A*. *citrulli* CFUs were quantified at 12, 15, and 18 days after inoculation as described above. The experiment was conducted three times, with six replicates per treatment.

### 4.5. Biofilm Formation Assay

Biofilm formation by the different strains and concentrations in three different media. The ability of *A*. *citrulli* strains to form biofilm was assessed qualitatively and quantitatively in three different media using 24-well and 96-well plates (Corning, Corning, NY, USA). After centrifuging, overnight cultures of *A. citrulli* strains were suspended in KB, M9, and M9-Fe^3+^ (12.5 mmol·L^−1^ Fe^3+^) media at two concentrations (OD_600_ = 0.3 and OD_600_ = 1.5). Each well was then loaded with 1 mL suspension for the 24-well plates and 100 µL for the 96-well plates. The plates were stationary cultured at 28 °C for 48 h. After heat fixation, each well was washed three times with phosphate buffered saline. Crystal violet (0.1%) was then added to each well and incubated for 30 min at 37 °C, after which the wells were washed with distilled water and then allowed to air dry. Biofilm formation for each strain was compared quantitatively by solubilizing the stained biofilms with 100% ethanol and measuring the OD_590_ with a microplate reader.

Biofilm formation under competition conditions. We also determined biofilm formation under competition conditions. The wells were loaded with mixed suspensions of *A*. *citrulli*:DH5α = 30:1 and *A*. *citrulli*: ddH_2_O = 30:1 and were cultured under the same conditions as the control. Three biological replicates were performed per *A*. *citrulli* strain per experiment, and the experiments were conducted three times.

### 4.6. Growth Ability Assay

To measure the growth ability of *A*. *citrulli* strains, cell suspensions at OD_600_ of 0.3 were diluted 10-fold and 100-fold in liquid KB, M9, and M9-Fe^3+^ (12.5 mmol·L^−1^ Fe^3+^) media, and then 200 µL of the suspensions were added to the wells of a 100-well plate. The plate was incubated at 28 °C with shaking at 220 rpm, and the OD_600_ was measured every 2 h for 96 h in a Bioscreen C chamber (FP-414 1100-C, Oy Growth Curves Ab Ltd., Helsinki, Finland). Three biological replicates were performed per *A*. *citrulli* strain per experiment, and the experiments were conducted three times.

### 4.7. Interspecies and Intraspecies Competition Assays

Interspecies competition assays. In interspecies competition with bacteria, the prey strains were *Escherichia coli* DH5α and *Pseudomonas aeruginosa* ATCC27853. Cells of killer and prey were mixed at a ratio of 20:1 (killer:prey) for *E. coli* DH5α and 30:1 for *P. aeruginosa* ATCC27853. Fungal prey strains were *Saccharomyces cerevisiae* Y2HGold and *Pichia pastoris* GS115, at a ratio of 20:1 (killer:prey).

Intraspecies competition assays. For the interspecies competition assays, cells of Aac5 and its derivative strains were mixed with the prey *A. citrulli* group I strain pslb65 at a ratio of 30:1 (killer:prey).

All the killer cultures (OD_600_ = 1.5) and prey cultures (OD_600_ = 2.0) were centrifuged and resuspended in sterile water. After co-culture, the mixtures were ten-fold serially diluted and spotted on plates containing antibiotics or selective medium. The survival bacterial populations of both killer and prey cells were quantified as log_10_CFU·mL^−1^.

### 4.8. Quantitative Reverse-Transcription (qRT-PCR)

Total RNA when T6SS was activated by cell density was extracted from *A. citrulli* Aac5, Δ*hcp*, and Δ*hcp*comp cultured in KB medium to OD_600_ = 0.3, 1.0, and 1.5. For in planta studies, the cotyledons were collected at 24, 48, and 72 h after injection at high concentration (10^6^ CFU·mL^−1^) for RNA extraction. RNA of the strains in the competition assays was extracted 3 h after co-culture with DH5α. All the RNA from strains under *in-vitro* conditions were extracted using the RNA prep pure Cell/Bacteria Kit (Tiangen, Beijing, China), while the RNA from strains grown under host conditions was extracted using TRIzol^®^ reagent (Invitrogen, Waltham, MA, USA). The qRT-PCR assay steps including genomic DNA removal, reverse transcription, and amplification were performed using the FastKing One Step RT-qPCR Kit (SYBR) on an Applied Biosystems 7500 Real-Time PCR System (ABI, Waltham, MA, USA). The primers used for qRT-PCR are listed in Appendix A. The *rpoB* gene was used as a reference for the normalization of gene expression. The relative expression of genes of interest was calculated using the 2^−ΔΔCT^ method [49]. Three replicates were used per experiment, and the experiment was conducted three times.

### 4.9. Western Blotting

Hcp protein expression was quantified by Western blotting to examine its secretion and activation. The strain Aac5-*hcp*-HA was generated from Aac5 by fusing an HA tag with the C-terminus of Hcp. The strain Δ*hcp*-pBBRMCS2-*hcp*-HA was used as the positive control due to its stable expression. All *A*. *citrulli* strains were cultured in KB broth to OD_600_ = 0.3, 1.0, and 1.5. To extract intracellular proteins, 1 mL of the bacterial suspension was centrifuged and resuspended in 200 μL of 4× Protein SDS PAGE Loading Buffer (TaKaRa, Kusatsu, Japan) and used as the whole-cell samples. For extracellular protein extraction, 30 mL of the bacterial suspension were centrifuged at 12,000× *g* and 4 °C for 3 min, and the supernatant was filtered through a 0.22 µm membrane filter to remove the cells. After precipitation with trichloroacetic acid (TCA) and eluting with acetone, the final pellets were resuspended in 200 μL of 4× Protein SDS PAGE Loading Buffer. All samples were heated at 95 °C for 10 min. The total proteins in the cell lysates and culture supernatants were collected from different strains co-cultured with *E*. *coli* DH5α for 3 h at a ratio of 30:1 to detect Hcp in competition assays. Hcp in the different strains cultured in M9 (with 0 mmol·L^−1^ Fe^3+^) and M9-Fe^3+^(12.5 mmol·L^−1^ Fe^3+^) was also detected in the cell lysates and culture supernatants to determine the effect of Fe^3+^ on Hcp. An antibody, Anti-HA-tag mAb-HRP-DirecT (MBL, Beijing, China) was used to detect the HA tag. The protein RpoB was used as the reference and detected by Direct-Blot™ HRP anti-*E*. *coli* RNA Polymerase β Antibody (Biolegend, San Diego, CA, USA).

### 4.10. Transcriptome Sequencing and Analysis under T6SS Activation Conditions

RNA (3 µg) from Aac5 and Δ*hcp* grown under cell density activated conditions of T6SS was extracted using RNA prep pure Cell/Bacteria Kit (Tiangen, Beijing, China) and used to construct the RNA sequencing libraries. The libraries were constructed and sequenced by Novogene Co., Ltd. (Beijing, China) as previously described [48]. The transcriptome data were analyzed by Novogene Co., Ltd. Differential gene expression between the WT strain and the *hcp* mutant strain (three biological replicates per strain) was analyzed using the DESeq2 R package (v1.20.0) [50] by Novogene Co., Ltd. (Beijing, China). Genes with an adjusted *p* value < 0.05 identified by DESeq were considered to be differentially expressed [51]. GO and KEGG pathway enrichments were performed as previously described by Guan et al. (2020) [48].

Secreted protein prediction was performed by SecretomeP-2.0 (https://services.healthtech.dtu.dk/service.php?SecretomeP-2.0, accessed on 27 April 2022) [38]. Domains were predicted by Pfam (http://pfam.xfam.org/, accessed on 27 April 2022) [39], and the homology of differentially expressed genes to effectors, immunity proteins, component proteins, regulators and accessory proteins recorded in SecReT6 was also predicted. (https://bioinfo-mml.sjtu.edu.cn/SecReT6/blast.php, accessed on 13 May 2022) [40].

### 4.11. Statistic Analysis

Statistical analyses were performed using one way ANOVA and Duncan’s new multiple range test in SPSS version 18.0 (SPSS Inc., Chicago, IL, USA). The statistical analyses of qRT-PCR were performed using independent sample *t*-tests. Differences were considered statistically significant at *p* values of 0.05 and 0.01. All charts were generated using GraphPad Prism 8.0 (GraphPad Software Inc., La Jolla, CA, USA).

## 5. Conclusions

By studying the function, expression, production and secretion of Hcp, we found that the T6SS in *A*. *citrulli* group II strain Aac5 affects various phenotypes related to colonization in watermelon, in-vitro growth ability, intra and inter-species competition, biofilm formation, and ferric iron absorption, and it can be activated by cell density, host condition and ferric irons. Based on transcriptome data, we found a few candidate effectors that need further identification.

## Figures and Tables

**Figure 1 ijms-23-09632-f001:**
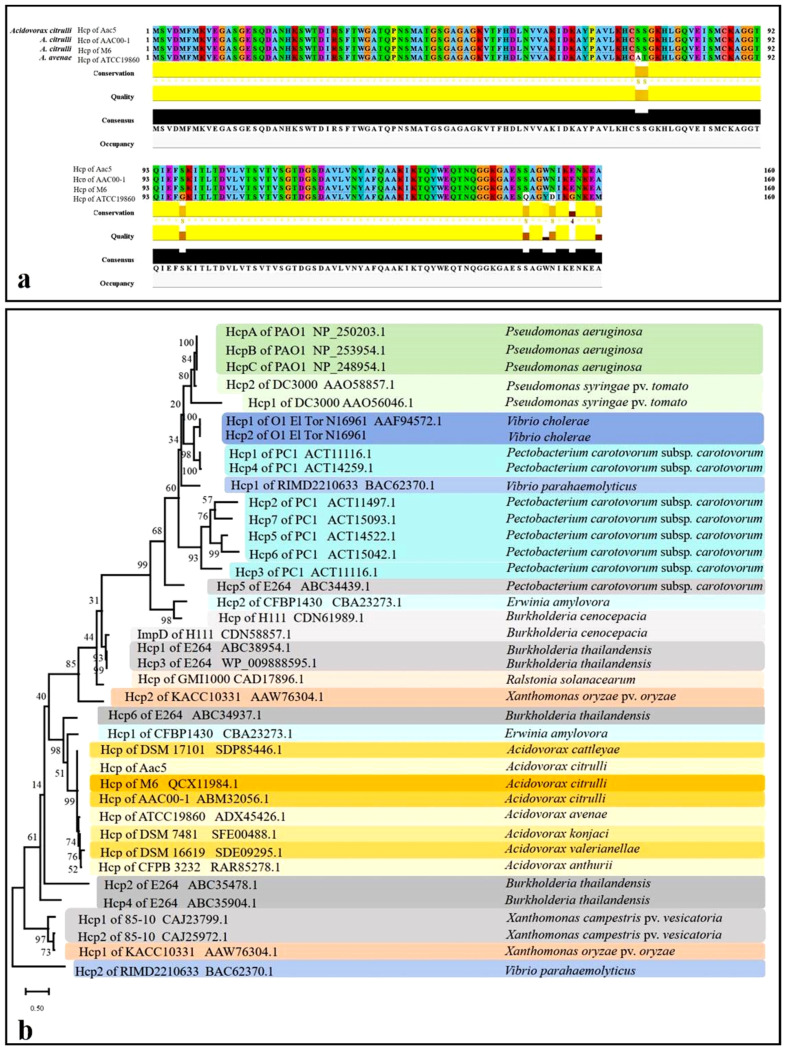
Sequence analysis of Hcp from *Acidovorax citrulli* Aac5. (**a**) Multiple sequence alignment of Hcp amino acid sequences from *A. citrulli* Aac5, AAC00-1, M6, and *A. avenae* ATCC19860. The alignment was performed using CLUSTALW2 and visualized in Jalview. (**b**) Maximum-likelihood phylogeny based on representative Hcp protein sequences. The evolutionary relationship was inferred in MEGA11 using the Maximum likelihood method and the JTT matrix-based model. The tree with the highest log likelihood (−8853.75) is shown. The bootstrap values are shown above the branches. Initial tree(s) for the heuristic search were obtained automatically by applying the Neighbor-Joining and BioNJ algorithms to a matrix of pairwise distances estimated using the JTT model and then selecting the topology with superior log-likelihood value. The tree is drawn to scale, with branch lengths measured in the number of substitutions per site. The tree was rooted with the outgroup Hcp2 of RIMD2210633 BAC62370.1, which has relatively low homology with other Hcps.

**Figure 2 ijms-23-09632-f002:**
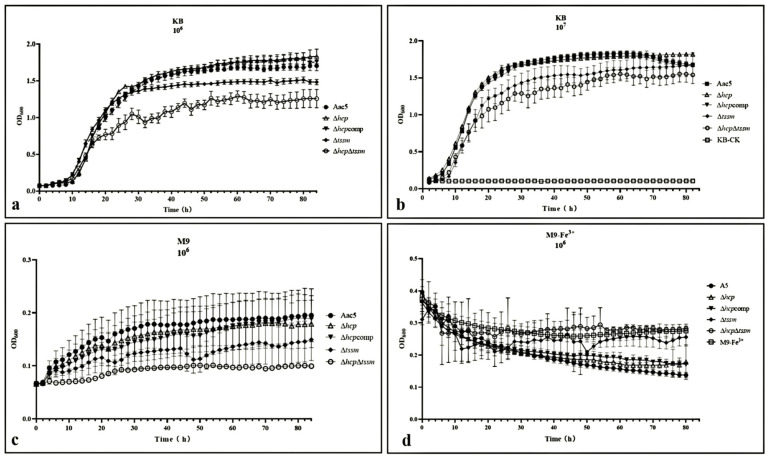
Growth ability of *Acidovorax citrulli* WT Aac5, Δ*hcp*, Δ*hcp*comp, Δ*tssm*, and Δ*hcp*Δ*tssm in vitro*. (**a**,**b**) Growth ability of the Aac5, Δ*hcp*, Δ*hcp*comp, Δ*tssm*, and Δ*hcp*Δ*tssm* strains in KB medium starting at 1 × 10^6^ CFU·mL^−1^ and 1 × 10^7^ CFU·mL^−1^, respectively. (**c**) Growth ability of Aac5, Δ*hcp*, Δ*hcp*comp, Δ*tssm*, and Δ*hcp*Δ*tssm* in M9 at an initial concentration of 1 × 10^6^ CFU·mL^−1^. (**d**) Growth ability of Aac5, Δ*hcp*, Δ*hcp*comp, Δ*tssm*, and Δ*hcp*Δ*tssm* in M9-Fe^3+^ (12.5 mmol·L^−1^ Fe^3+^) at an initial concentration of 10^6^ CFU·mL^−1^. Cell suspensions of each strain grown to a density of OD_600_ of 0.3 were diluted 10-fold and 100-fold in liquid KB, M9, and M9-Fe^3+^ media, and then 200 µL of the suspensions were added to wells of a 100-well plate. Three biological replicates were performed per *A*. *citrulli* strain per experiment and the experiment was conducted three times. The plate was incubated at 28 °C with shaking at 220 rpm/min, and the OD_600_ was measured every 2 h for 96 h in a Bioscreen C chamber.

**Figure 3 ijms-23-09632-f003:**
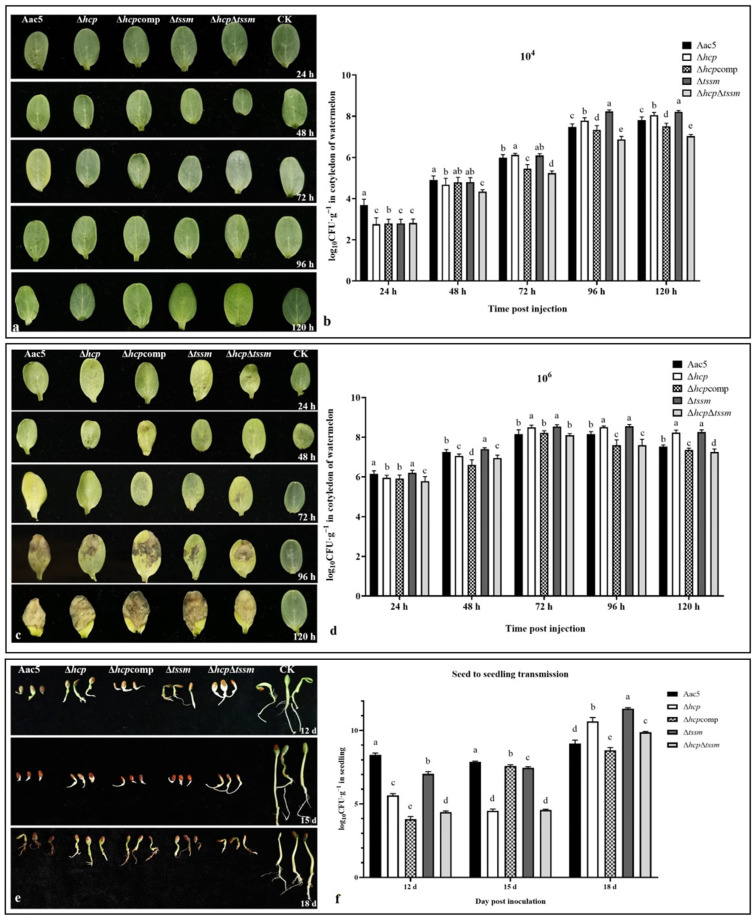
Virulence and colonization of *Acidovorax*
*citrulli* strains Aac5, Δ*hcp*, Δ*hcp*comp, Δ*tssm*, and Δ*hcp*Δt*ssm*. (**a**) Symptoms caused by inoculation with WT, Δ*hcp*, Δ*hcp*comp, Δ*tssm*, Δ*hcp*Δ*tssm*, and sterile water (CK) at 24, 48, 72, 96, and 120 h post inoculation (hpi). The strains were cultured until the logarithmic phase, washed, resuspended in sterile water at 1 × 10^4^ CFU·mL^−1^, and injected into the watermelon cotyledons. (**b**) Bacterial population levels in watermelon cotyledons. Significant differences were found between different letter substitutes on the column calculated by Duncan’s new multiple range test (*p* = 0.05). The strains were cultured until the logarithmic phase, resuspended in sterile water at 1 × 10^4^ CFU·mL^−1^, and injected into the watermelon cotyledons. (**c**) Symptoms caused by WT, Δ*tss**m*, Δ*hcp*Δ*tssm*, Δ*hcp*, Δ*hcp*comp, and sterile water (CK). The strains were resuspended in sterile water at 1 × 10^6^ CFU·mL^−1^ and injected into the watermelon cotyledons. (**d**) Bacterial population levels in watermelon cotyledons (1 × 10^6^ CFU·mL^−1^). Significant differences were found between different letter substitutes on the column calculated by Duncan’s new multiple range test (*p* = 0.05). (**e**) Disease symptoms on germinating watermelon seeds at 12, 15, and 18 d. The strains were cultured to OD_600_ = 0.3 (3 × 10^8^ CFU·mL^−1^), washed, and resuspended in sterile water. Seeds soaked in bacterial suspensions for 4 h were transferred to 15 mL centrifuge tubes containing sterile absorbent cotton, filter paper, and sterile water. (**f**) Bacterial population levels on germinating watermelon seedlings in the seed-to-seedling transmission assay. Significant differences were found between different letter substitutes on the column calculated by Duncan’s new multiple range test (*p* = 0.01).

**Figure 4 ijms-23-09632-f004:**
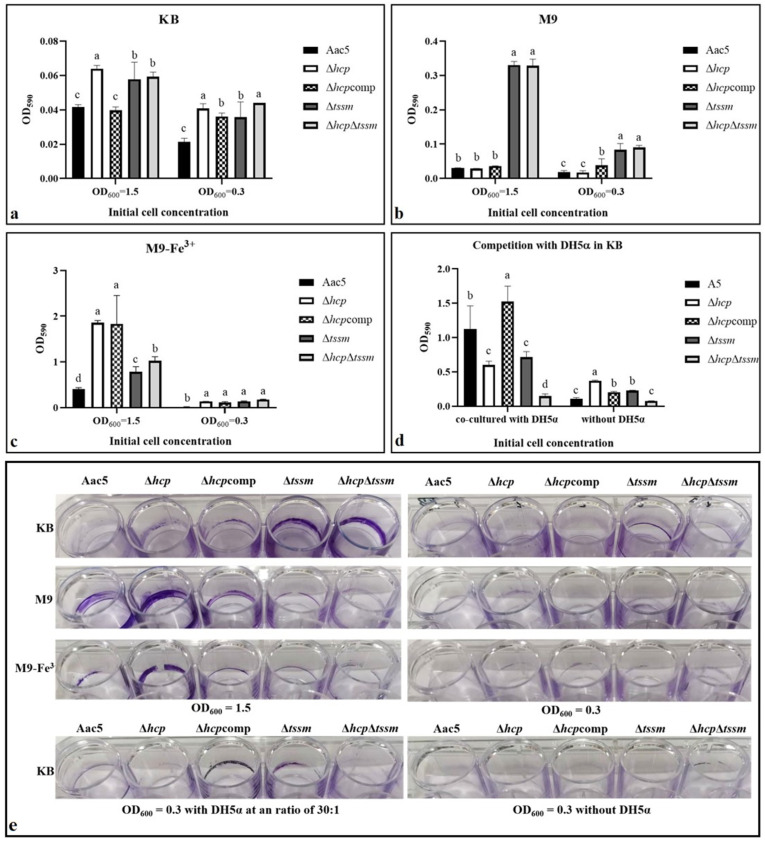
Biofilm formation of *Acidovorax citrulli* strain Aac5, Δ*hcp*, Δ*hcp*comp, Δ*tssm*, and Δ*hcp*Δt*ssm* grown on three different media, and under competition conditions. (**a**–**c**) Quantification of biofilm formation of cells grown in KB, M9, and M9-Fe^3+^ (12.5 mmol·L^−1^) media at an initial cell concentration of OD_600_ = 0.3 and 1.5 by measuring absorbance at OD_590_. (**d**) Biofilm formation of cells grown in KB with *E*. *coli* DH5α at an ratio of 30:1. (**e**) Qualitative visualization of biofilm formation of strains grown on KB, M9, and M9-Fe^3+^ media at an initial cell concentration of OD_600_ = 0.3 and 1.5, and under competition conditions with *E*. *coli* DH5α. Significant differences were found between different letter substitutes on the column calculated by Duncan’s new multiple range test (*p* = 0.05).

**Figure 5 ijms-23-09632-f005:**
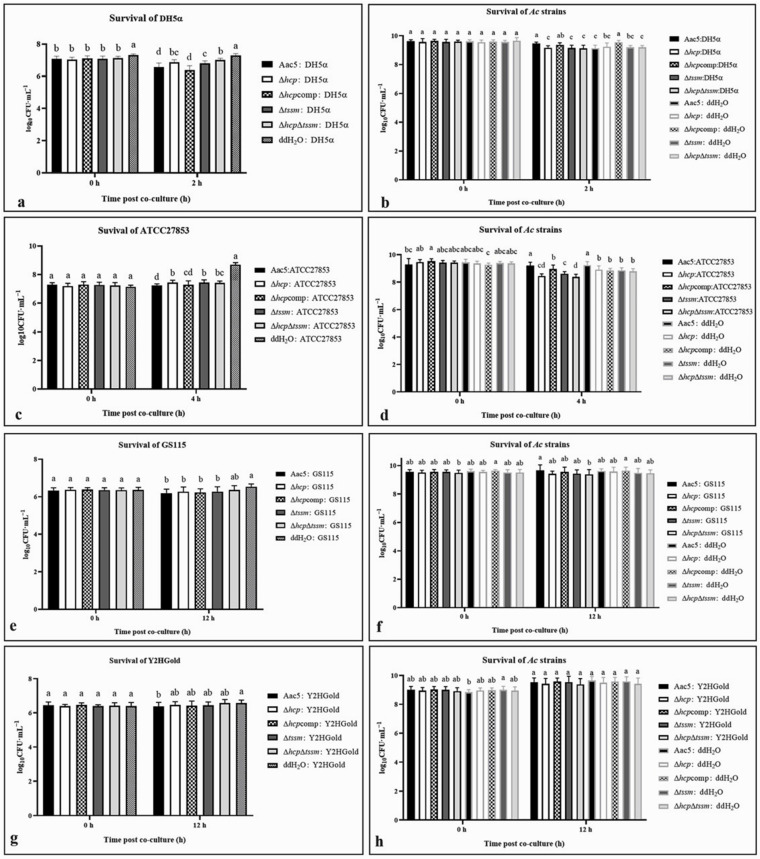
Interspecies competition of *Acidovorax citrulli* strains (*Ac* strains for short) Aac5 (WT), Δ*hcp*, Δ*hcp*comp, Δ*tssm*, and Δ*hcp*Δ*tssm* with *Escherichia coli* DH5α-pBBRMCS5, *Pseudomonas aeruginosa* ATCC27853, *Saccharomyces cerevisiae* Y2HGold, and *Pichia pastoris* GS115. (**a**) Survival of *E*. *coli* DH5α-pBBRMCS5 on LB amended with gentamicin (Gm) after 2 h of co-culture with *A. citrulli* killer strains (Aac5, Δ*hcp*, Δ*hcp*comp, Δ*tssm*, and Δ*hcp*Δ*tssm*) at a killer:prey ratio of 20:1. (**b**) Survival of Aac5, Δ*hcp*, Δ*hcp*comp, Δ*tssm*, and Δ*hcp*Δ*tssm* strains on KB amended with Amp and Km after 2 h of co-culture with DH5α-pBBRMCS5. (**c**) Survival of *P*. *aeruginosa* ATCC27853 on CFC selective agar after 4 h of co-culture with the killer strains (Aac5, Δ*hcp*, Δ*hcp*comp, Δ*tssm*, and Δ*hcp*Δ*tssm*) at a killer:prey ratio of 30:1. (**d**) Survival of Aac5, Δ*hcp*, Δ*hcp*comp, Δ*tssm*, and Δ*hcp*Δ*tssm* on KB amended with Amp and Km after 4 h of co-culture with ATCC27853. (**e**) Survival of *S*. *cerevisiae* Y2HGold on YPDA agar amended with Gm after 12 h of co-culture with killer strains (Aac5, Δ*hcp*, Δ*hcp*comp, Δtssm, and Δ*hcp*Δ*tssm*) at a killer:prey ratio of 30:1. (**f**) Survival of Aac5, Δ*hcp*, Δ*hcp*comp, Δ*tssm*, and Δ*hcp*Δ*tssm* on KB amended with Amp, Km, and AbA after 12 h of co-culture with Y2HGold. (**g**) Survival of *Pichia pastoris* GS115 on YPDA medium amended with Gm after 12 h of co-culture with killer strains (Aac5, Δ*hcp*, Δ*hcp*comp, Δ*tssm* and Δ*hcp*Δ*tssm*) at a killer:prey ratio of 30:1. (**h**) Survival of Aac5, Δ*hcp*, Δ*hcp*comp, Δ*tssm*, and Δ*hcp*Δ*tssm* on KB medium amended with Amp, Km, and AbA after 12 h of co-culture with GS115. All strains were cultured to an OD_600_ = 1.5 and co-cultured in 24-well-plates with ddH_2_O. After co-culture, the mixtures were ten-fold serially diluted and spotted onto plates with antibiotics or selective media. The survival of both killer and prey cells was quantified by colony count and converted to log_10_CFU·mL^−1^. Significant differences were found between different letter substitutes on the column calculated by Duncan’s new multiple range test (*p* = 0.05).

**Figure 6 ijms-23-09632-f006:**
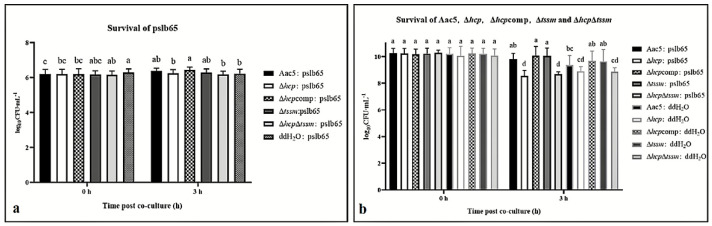
Intraspecies competition between *Acidovorax citrulli* group I strain pslb65-pBBRMCS5 and group II strains Aac5, Δ*hcp*, Δ*hcp*comp, Δ*tssm*, and Δ*hcp*Δ*tssm*. (**a**) Survival of pslb65-pBBRMCS5 in KB medium amended with Ampicillin (Amp), and Gentamicin (Gm) after 3 h of co-culture with the killer strains (Aac5, Δ*hcp*, Δ*hcp*comp, Δ*tssm*, and Δ*hcp*Δ*tssm*) at a killer:prey ratio of 30:1. (**b**) Survival of Aac5, Δ*hcp*, Δ*hcp*comp, Δ*tssm*, and Δ*hcp*Δ*tssm* in KB amended with Amp and Km after 3 h of co-culture with pslb65-pBBRMCS5. All strains were cultured to OD_600_ = 1.5 and co-cultured in 24-well-plates with ddH_2_O. After co-culture, the mixtures were ten-fold serially diluted and spotted onto plates amended with antibiotics. The survival of both killer and prey cells was quantified by colony count and converted to log_10_CFU·mL^−1^. Significant differences were found between different letter substitutes on the column calculated by Duncan’s new multiple range test (*p* = 0.01).

**Figure 7 ijms-23-09632-f007:**
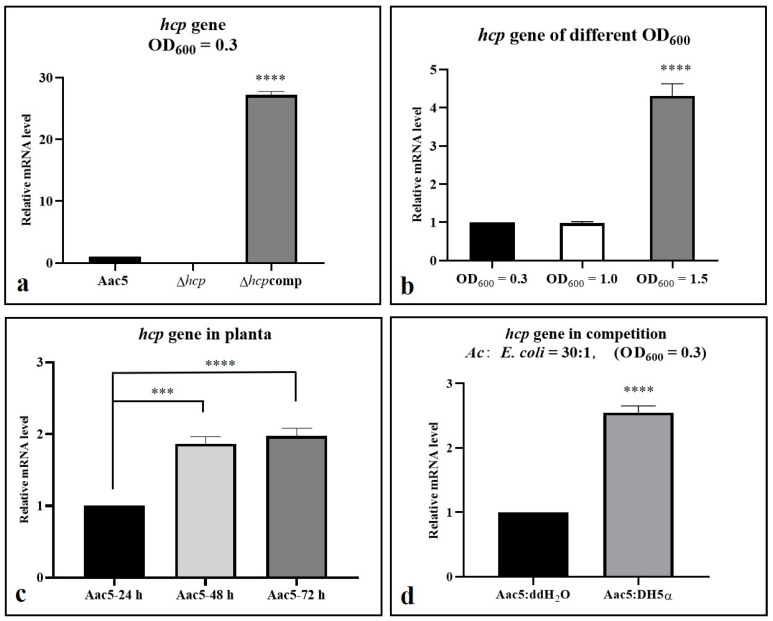
Activation of *Acidovorax citrulli hcp* as determined by *hcp* relative expression level. qRT-PCR assays were performed to determine the relative *hcp* mRNA levels under different growth conditions. (**a**) Relative expression of *hcp* in Aac5, Δ*hcp*, and Δ*hcp*comp under inactivated conditions (OD_600_ = 0.3). (**b**) Relative expression of *hcp* in WT strain Aac5 at different cell concentrations (OD_600_ = 0.3, 1.0, and 1.5). (**c**) Relative expression of *hcp* in Aac5 at 24, 48, and 72 h post inoculation (hpi) of watermelon cotyledons (10^4^ CFU·mL^−1^). (**d**) Relative expression of *hcp* in Aac5 and Δ*hcp*comp cells co-cultured with *E. coli* DH5α at a ratio of 30:1 for 3 h. The relative expression levels of *hcp* were calculated using the 2^−ΔΔCT^ method. The data were analyzed by independent sample *t* test. “***” means *p* value < 0.001 and “****” means *p* value < 0.0001.

**Figure 8 ijms-23-09632-f008:**
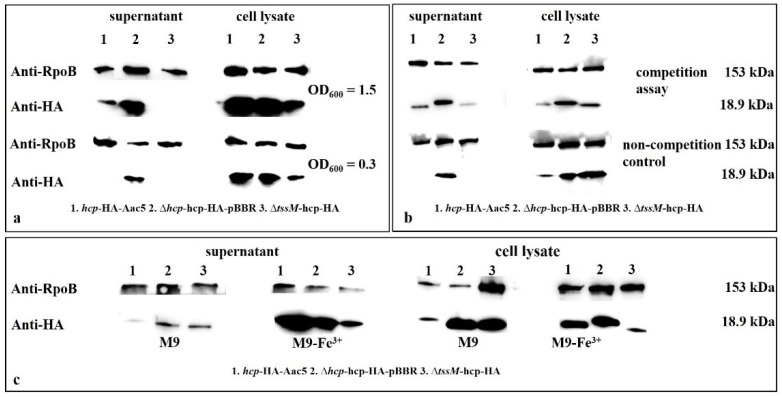
Production and secretion of Hcp detected by Western blotting of Hcp-HA. (**a**) Hcp in *hcp*-HA-Aac5, and Δ*hcp*-*hcp*-HA-pBBR, Δ*tssm*-*hcp*-HA at two cell densities (OD_600_ = 0.3 and 1.5) were detected in the cell lysates and culture supernatants by Western blotting. (**b**) Hcp in competition assays was detected by Western blotting; the total protein in the cell lysates and culture supernatants was collected from different strains co-cultured with *E. coli* DH5α for 3 h at a ratio of 30:1. (**c**) Hcp in *hcp*-HA-Aac5, Δ*hcp*-*hcp*-HA-pBBR, and Δ*tssm-hcp*-HA cultured in M9 (with 0 mmol·L^−1^ Fe^3+^) and M9-Fe^3+^ (12.5 mmol·L^−1^ Fe^3+^) was detected in the cell lysates and culture supernatants by Western blotting; HCP in *hcp*-HA-Aac5 was fused with an HA tag at the C-terminus. Δ*hcp*-*hcp*-HA-pBBR was used as the positive control because the expression of Hcp was constant. RpoB protein was used as the internal control for normalization of gene expression.

**Figure 9 ijms-23-09632-f009:**
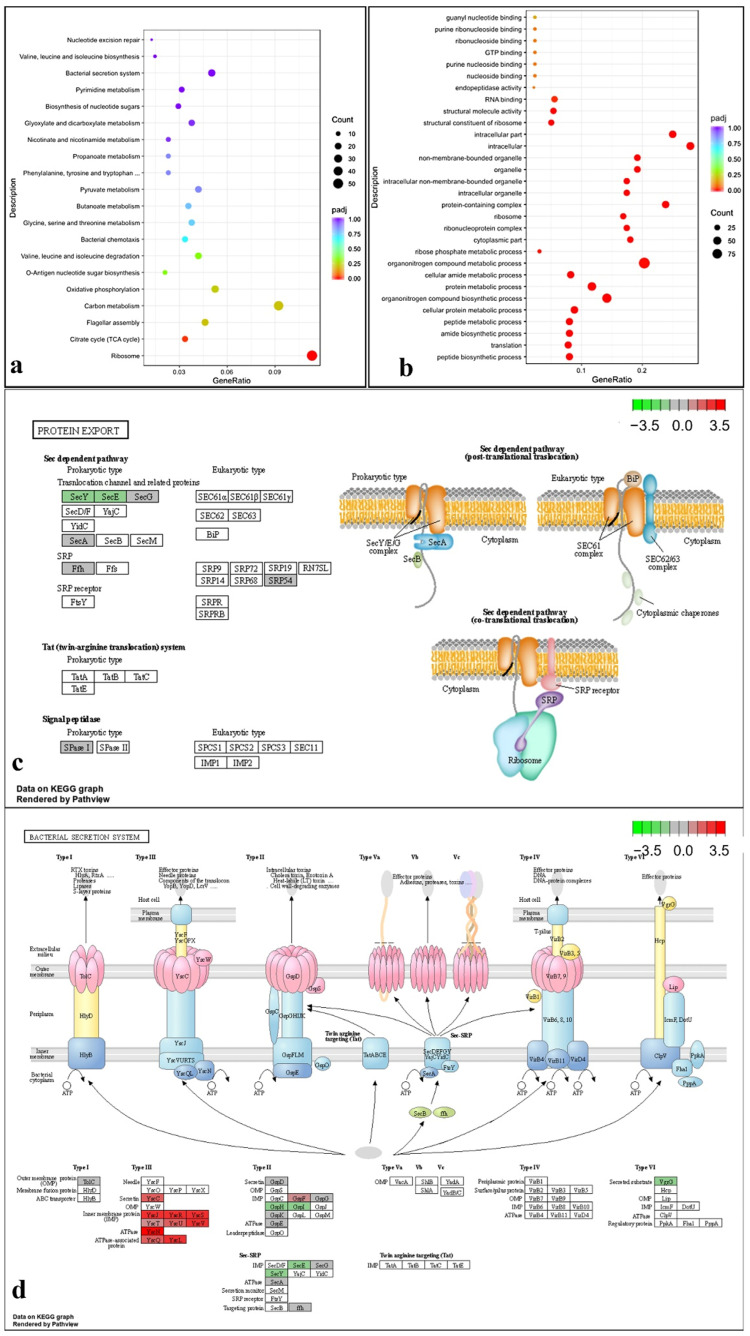
Functional classification and enrichment of genes that showed differential expression between the *hcp* mutant and the WT strain of *Acidovorax citrulli* by (**a**) KEGG (Kyoto Encyclopedia of Genes and Genomes) and (**b**) GO (Gene Ontology) analyses. The expression of some genes for protein export (**c**) and secretion systems (**d**) in KEGG pathways were affected.

## Data Availability

Not applicable.

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
