# Peer review of "Hcp of the Type VI Secretion System (T6SS) in Acidovorax citrulli Group II Strain Aac5 Has a Dual Role as a Core Structural Protein and an Effector Protein in Colonization, Growth Ability, Competition, Biofilm Formation, and Ferric Iron Absorption"

_ijms, 2022, doi:10.3390/ijms23179632_

Round 1
Reviewer 1 Report
The manuscript of Fei et al. provides new insights into the type VI SS of Acidovorax citrulli Group II, an important bacterial phytopathogen.
The study is well planned, performed and written, and provides new and useful data on a basic component of the virulence machinery of the bacterium.
Since there are two groups of the pathogen, it is importante to add the host plants from where have ben isolated the four strains the Authors worked with, namely M6 and ps1b65 for the group I, and AAC-00-1 and KACC17005 for the group II.
In addition, the Authors should explain why they changed the bacterial concentration in their virulence and colonization assays (from 104 CFU/ml to 106 CFU/ml).
Author Response
Response to Reviewer 1
Dear reviewer:
We feel great thanks for your professional review work on our article. As you are concerned, there are several problems that need to be addressed. According to your nice suggestions, we have made extensive corrections to our previous draft. The detailed corrections are listed below. All the numbers of pages and lines refer to the revised manuscript file.
The manuscript of Fei et al. provides new insights into the type VI SS of Acidovorax citrulli Group II, an important bacterial phytopathogen.
The study is well planned, performed and written, and provides new and useful data on a basic component of the virulence machinery of the bacterium.
Since there are two groups of the pathogen, it is important to add the host plants from where have been isolated the four strains the Authors worked with, namely M6 and ps1b65 for the group I, and AAC-00-1 and KACC17005 for the group II.
Response: Thank you very much for your advice. The information of the host plants of the four strains has been added in the revised manuscript as the comments (lines 155-163). The revised content is as follow: “Based on the genome data on NCBI, we chose the whole T6SS gene clusters predicted in the genome of four A. citrulli representative strains (group II strains AAC00-1 which was isolated from Citrullus lanatus var. citroides, KACC17005 isolated from Citrullus lanatus; group I strains M6 and pslb65 isolated from Cucumis melo) to conducted sequence alignment.”
In addition, the Authors should explain why they changed the bacterial concentration in their virulence and colonization assays (from 104 CFU/ml to 106 CFU/ml).
Response: Thank you very much for your advice. We used the high concentration of 106 CFU·mL-1 to observe the symptom and analyze the bacterial population level of stationary phase and decline phase. But the bacterial population growth too fast to analyze. So the low concentration of 104 CFU·mL-1 was used to study colonization ability and the bacterial population level before stationary phase. When the concentration was 104 CFU·mL-1, the disease severity caused by Acidovorax citrulli was mild.
The revised content is as follows (lines 257-262): “When the concentration of bacterial suspension was 104 CFU·mL-1, the cotyledons showed only mild symptoms (disease severity was about 1 to 3) [36, 37] and the bacterial population was in growth stage. When the concentration was 106 CFU·mL-1, the bacterial population started to decrease after 72 h. In our virulence assays, deletion of hcp and tssm did not significantly affect the disease severity on cotyledons when the initial concentrations were 104 CFU·mL-1 and 106 CFU·mL-1 (data not shown, p = 0.05).”
If there are any other modifications we could make, we would like very much to modify them and we really appreciate your help. Thank you very much for your help.
Sincerely yours,
Yuwen Yang, Tingchang Zhao and Nuoya Fei
State Key Laboratory for Biology of Plant Diseases and Insect Pests, Institute of Plant Protection, Chinese Academy of Agricultural Sciences
China, Beijing, Yuanmingyuan, 2
100193 Beijing
China
Email: yangyuwen@126.com, zhaotgcg@163.com, fei_nuoya@126.com
Reviewer 2 Report
Authors presented research to investigate the role of Hcp of the T6SS in Acidovorax citrulli group 2 strains. The study was largely inspired by work done in Pseudomonas aeruginosa by Silverman et al. (2013). The hypothesis for the study is sound and should improve knowledge of T6SS in general and in A. citrulli. However, authors need to clarify certain experiments and explain their choice. I also think authors need to narrow their claims and only explain what they see from the experiment. A major concern in some earlier conclusions arose from the fact that authors did not first carry out in planta assays to understand strain growth patterns and what days after inoculation may approximate to populations of OD600= 1.5 at which T6SS is activated for WT and mutant strains. The log CFU values provided at 120h and 18 in Figure 1 for example (which were inoculated at OD600=0.3), could be shown alongside the approximate log CFU values of bacterial population at OD600= 1.5 to see if the population are comparable, or perhaps these effects were not due to T6SS at all if the bacterial population did not reach OD600=1.5 in planta within the days compared. Also, no graph or table for transcriptomics comparison was provided. I also cannot find Table S3 as stated by the authors. The discussion was difficult to follow, and authors need to refer to relevant sections in the paper while discussing. Because of the importance of this work, I am going to suggest acceptance with minor revision, hoping that the authors can address these points as soon as possible. Perhaps authors have better explanation that they wish to provide. I made relevant suggestions and I look forward to seeing an improved version and author’s response. Below are additional comments:
Line 15: Include a period (.) between “Acidovorax citrulli” and “Research”
Line 21: The full meaning of hcp was earlier given but not tssm. Authors can provide the meaning in brackets or writing it out in full.
Line 48: change “amount of T6SS” to “number of T6SS”
Line 50: change “and also studied” to “and is studied”
Line 53: The statement “There are six sets of T6SS gene cluster in B. thailandensis” should come before the immediate previous sentence because the previous sentence compares T6SS in citrulli with B. thailandensis.
Line 76-77: need to be reworded for flow. Throwing TssM in the beginning of the sentence brings an immediate break that is not in tandem with the previous sentence. Suggestion: “Additionally, since TssM is a key component of the T6SS apparatus (16) and present in the middle of A. citruli T6SS gene cluster (20), we used a tssm mutant as a T6SS deficient control.”
Line 85: “In a preliminary study”
Line 71 -98: I suggest reorganization of this section thus:
“Hcp (Hemolysin co-regulated protein, also known as TssD) is both a special effector and a structural member of the inner tube of the T6SS [25]. Because of the secretory function and the assembly of the secretory channel with VgrG (Valine-glycine repeat 70 protein G; TssI) and PAAR (Pro-Ala-Ala-Arg-repeat-containing protein) [26, 27, 28], Hcp and VgrG can be used to study T6SS activation [29, 30, 31, 32, 33]. In a preliminary study, we found that the expression of one of the T6SS vgrG genes Aave_2840 was up-regulated when T3SS gene hrpE was absent in the A. citrulli group II strain Aac5 (unpublished transcriptome data). When Type 4 pilus (T4P) gene pilA was deleted, the expression of both hcp and vgrG was affected (unpublished transcriptome data). These unexpected data of T6SS gene indicate that T6SS of A. citrulli may have multiple biological functions, which needs to be proven.
At present, there is no report of the T6SS activated condition in A. citrulli, which is needed for the further research such as effector identification and transcription regulation of T6SS. In addition, due to the substantial differences between the two groups of A. citrulli strains and the horizontal transfer of T6SS genes [34, 35], the function of the T6SS in A. citrulli may vary by groups and strains. T6SS may also be related to other biochemical processed to realize its multiple functions.
In this study, we investigated the role of the key element hcp in virulence, competition, biofilm formation, growth ability and ferric ion absorption to better understand T6SS in A. citrulli. Because there are 12 VgrG but one Hcp in A. citrulli, we select Hcp as a marker for studying T6SS activation and secretion in this study. The transcriptional orientation of the hcp gene differs from that of all the other genes in the T6SS cluster of A. citrulli, which makes the hcp gene unique. Additionally, since TssM is a key component of the T6SS apparatus (16) and present in the middle of A. citruli T6SS gene cluster (20), we used a tssm mutant as a T6SS deficient control. The activation of hcp at different conditions was 93 determined by qPCR and western blotting to determine the activation condition of T6SS 94 in A. citrulli strain Aac5. In addition, the transcriptome data under the T6SS-activated condition was analyzed to study the possible relation of T6SS and other system and to find candidate effectors. These results will support further research on potential T6SS 97 effectors and the interaction between Hcp and VgrG proteins in A. citrulli. "
Line 103: Capitalize “We”
Comment on Figure S1: Figure S1 is not labelled in the supplementary file. However, I assume the alignments before Figure S2 is Figure S1. It would be clearer if you place some pointers to the places of single mutations for each of the genes. It seems this is shown in fha but not others.
Line 107-109: Please be consistent. The locus tag of A. citrulli group II Aac5 was not mentioned while the locus tags for AAC00-1 and M6 were mentioned.
Comment on Figure 1: The figures appear blurred and could be improved.
Figure 1a: Some of the words do not appear in full: Citrulli appears as citrull in some instances. Same for avenae.
Figure 1b: This tree does not seem to be rooted with an outgroup, so it is difficult to say “A. citrulli and Hcp of T6SS-2, 4, and 6 of B. thailandensis were clustered in the same clade” as stated in line 113. This is especially because the bootstrap support for where the Hcp of citrulli strains diverged from the B. thailandensis is either 82 or 97, which suggests a strong support for divergence, not clustering. Please, root the tree with any of the species and see what you get.
Line 144: From Figure 2b, this statement is confusing:
“In the first 24 h, Δtssm, ΔhcpΔtssm, Δhcp, and the Hcp-complemented strain Δhcpcomp barely grew in the host tissue, while only the WT strain Aac5 maintained the initial cell concentration”
Figure 2b shows that the populations of Δtssm, ΔhcpΔtssm, Δhcp, and the Hcp-complemented strain Δhcpcomp in planta reduced, compared to the WT which maintained its initial inoculated population. So saying “barely grew in the host tissue” could be misleading.
Comment regarding Figure 2b: it is interesting that at 96h and 120h, Δtssm has higher population than all others.
Line 159-162: The conclusions from the virulence experiments, which should be correctly differentiated into in planta population assay and virulence assay, need to be improved. The authors concluded from their experiment that strain Aac overcame the loss of virulence by increasing the cell numbers when either hcp or tssm was deleted, and that the mechanism was not effective when both hcp and tssm were deleted. This statement is too general in my opinion.
First, I expected the authors to delineate between in planta population growth assays and virulence, as they do not always mean the same (Please see Tardy et al. 2019: https://www.pnas.org/doi/pdf/10.1073/pnas.1901556116). The authors should also be familiar with this as they have demonstrated in their previous paper where they carried out different experiments for virulence assays and in planta population growth (See Guan et al. 2020; and Wang et al. 2022: https://www.frontiersin.org/articles/10.3389/fmicb.2021.783862/full). However, this is not clearly differentiated in this paper.
Secondly, from the results presented, the statement does not appear to be true. Figure 2a does not show any differences in virulence in my opinion. In Figure 2b, there was only reduced bacterial population in planta within 24 h when the initial inoculum was 104 CFU/ml. Subsequently, after 24 h, all strains increased their populations at different rates. The in planta bacterial population at 106 CFU/ml at 24 h is comparable between WT and Δtssm strains, while all others also increased as discussed by the authors (Figure 2d). In figure 2c, the pattern of change in virulence does not suggest that there was loss of virulence in the first place. Necrotic symptoms only appeared on the WT at 96 hpi, whereas it appears that necrosis development already began earlier on some of the other mutant strains at 72 hpi. Regardless, there was necrosis on all (except the CK) at 96h. At 120 h, it is difficult to differentiate between the level of necrosis.
I suggest using the virulence assay rating scale as in Guan et al. 2020 to approximate the ratings as they appear in figures 2a and c to improve this discussion.
Line 161-163: Concerning the statement of the authors: “Another possible explanation could be that when hcp or tssm were deleted separately, Aac5 lost the ability to control cell density.” This statement is too wild and does not appear to follow from the result presented. While there appears to be continued increase in bacteria population for strains Δtssm and Δhcp at 120 hpi for the initial inoculation concentration of 104 CFU/ml, this is not case for the 106 CFU/ml inoculation. In Figure 2d, at 120 hpi, the in planta population for strains Δtssm and Δhcp are lower than at 72 and 96 hpi. So, the results do not necessarily show loss of ability to control cell density.
In the seed-to-seedling transmission assay, it would be important to take the bacterial population within 24 h of transferring to the 15 ml centrifuge tube as was done in the in planta population growth assay. I understand that the authors stated that the experiment was repeated three times, with 6 replicates per treatment, which should generally be robust. However, by not sampling bacterial population within 24 h, the authors cannot establish the initial population on the seeds, and it affects the inference that can be derived from this important experiment. From the previous section (figure 2d), although it was on cotyledons, the evidence suggests that at higher cfu/ml, there was comparable bacterial population for WT and Δtssm strain; similarly for Δhcp and Δhcpcomp at 24 h. Given that these strains were generated in this study, it makes sense to provide this information. Readers would not be sure what to expect on seeds, with respect to the initial populations of these strains on seeds. It is therefore difficult to infer anything from the drastic changes in bacterial populations within 3 days of each other as presented in Figure 2f.
Perhaps authors already have this result or have convincing explanation. Otherwise, I suggest repeating this experiment and replicated at least three times, and sampling bacterial populations within 24h and 6dpi, including the other times used by authors.
Line 188-189: I think it is interesting to state that WT and Δhcpcomp behaved differently at OD 1.5 and 0.3
Line 192: “When co-cultured with DH5α, all strains formed more biofilm compared to without DH5α”: I do not see the well with DH5α alone. Or what was the control as suggested in line 551? Also check line 550: is it supposed to be ddH2O:DH5α = 30:1 or A. citrulli: ddH2O = 30:1?
Growth ability assay: This section should have come much earlier in the experiment and as Figure 2. This could have helped to buttress the statement made on line 161-163 regarding cell density regulation, (however I still think my earlier statement remain valid for that section, especially when Δtssm is still showing different result).
Line 224-
Figure 5: Please clearly state what “Ac” means in the description of the figure for b, d, f, h. Is it Ac or Aac5? For the other figures (a, c, e, g), authors used strain name. it is also difficult to differentiate what the bar colors stand for, since a single color is used to represent multiple phenotypes. I understand the difficultly because of multiple comparison. However, authors have not used the full range of colors available. Some of the colors used in Figure 3 (and more) could be used to differentiate between the bars.
Figure 6a legend on the main graph: Aac5:pslb65, not A5. Same for Figure 6b legend on the main graph: the second Aac5 is written as A5. This figure has more colors representing different bars. However, two colors are used for two phenotypes. Authors need to state/explain using similar color for strains:pslb65 and their corresponding co-culture with ddH2O in Figure 6b.
Line 240-244, Comment on Figure 6a: At 0h post co-culture (pcc), WT ddH20:pslb65 (a) already has significant difference in colony count than Aac5:pslb65 (c). It is therefore interesting that after 3h, they have comparable colony counts. This appears to be the same scenario with Δhcp and Δhcpcomp. It might be informative to include comparison of each treatment at 0h and 3h, with respect to the control ddH20:pslb65. By doing this, the differences in co-culture between the WT Aac5 and the various mutants would become clearer.
Line 288: This sentence is hanging: “To explore Hcp activation conditions, the following factors were examined.” What factors?
Implications of the qRT-PCR result on earlier conclusions
The results from qRT-PCR put some of the earlier conclusions to question. From the qRT-PCR experiments, we see that T6SS was not activated at OD600 of 0.8. Meanwhile, experiments in 4.4 (in planta population assay, virulence and seed-to-seedling transmission) were done at OD600= 0.3. This puts further dent on the conclusions made in lines 159-163 since it is not clear that the T6SS was activated within the first 24 h of inoculation. Is it wise to compare at 24 hpi? It also makes the discussion on seed-to-seedling transmission difficult to agree with, especially since we don’t know how the variation in populations began in the early days and perhaps, TS66 may have been differentially activated in bacterial populations. This is same with growth ability assay and the previous in vitro assays.
The qPCR experiments also focused on the strains grown under KB medium (without antibiotic resistance) (4.8). In 4.1, the authors stated that media used for experiments had antibiotic resistance. Given that many of these experiments were carried out at OD=0.3, it would be important to see the OD at which T6SS is activated under the various antibiotic resistance conditions where these experiments were carried out. Antibiotics have effects on many aspects of bacterial growth, including biofilm formation (https://www.frontiersin.org/articles/10.3389/fmicb.2020.02109/full), secretion systems (https://www.sciencedirect.com/science/article/pii/S0167488914000512), etc. Perhaps, this will provide answers to some of the differences (or no differences) seen in the populations in the different experiments. Perhaps, T6SS was not activated at 0 h at OD OD=0.3 in Figures 2-5.
Comment on Section 2.8: The authors need to include a supplementary figure showing comparable variations in gene regulations under the various conditions. Figure 9 as presented shows nothing in that respect. Authors stated that vgrG in Δhcp was downregulated, but I could not find the results where these were compared. I also cannot find Table S3 in the files available to me. This applies to most conclusions in this section.
Line 351, Figure 9 legend: remove the first (c ) and (d)
Comments on discussion:
This section needs to be thoroughly organized so that paragraphs address major points at a time. The discussion on many results are packed together in a single paragraph which makes it difficult to follow. Also, authors should reference the particular figures addressed in the discussion. I believe some of the discussion can also be improved after the earlier comments are addressed.
Line 360: Prefix TssM with “Deletion of tssm, the gene responsible for…”
Line 576-577: No mention of plant before throwing this sentence: “The cotyledons were collected at 24, 48, and 72 h after injection at high concentration (106 CFU·ml−1) for RNA extraction.”. Add: “For in planta studies” or variation of author’s choice.
Line 398-399: Authors stated “ Therefore, Hcp and the T6SS might not be activated under this growth condition.”. So why did you use lower ODs for Figure 2 etc if you knew this? The authors further stated:
“It is critical to define the conditions necessary for T6SS activation for further research, such as screening, identifying, and characterizing T6SS effectors”. I agree.
Line 406: Which figure: “The cell density at 72 hpi in the host had not reached the same level as that of OD600=1.5 in KB medium…”
Line 412-413: I am not sure you can broadly say these: “Based on the above results, T6SS and its key element Hcp in A. citrulli are involved in watermelon cotyledon colonization and seed-to-seedling transmission, interspecies and intraspecies competition, and regulation of cell density.” Please see or address earlier comments on Figure 2.
Line 415: Include figure: “but in Aac5 we found that the production and secretion of Hcp was promoted by ferric iron (Figure?).
Line 422-442: Authors need to provide a table or graph comparing transcriptomics results.
Line 428-430: I can’t see this comparison: “Among the differentially expressed genes under the cell density activated condition, we found twelve genes encoding secreted proteins predicted by SecretomeP-2.0.”
Line 432-463: Authors need to stop starting sentences with conjunctions such “But”, “And”:
“But further experiments such as secretion assay and function analyses need to be conducted to identify it as a T6SS effector. And there were five genes encoding secreted proteins that showed certain homology to effectors proteins recorded in SecReT6. But the homogeneity of some genes was not high enough, they also need further experiments of identification.”
Line 425-426: where can I see this data and why did authors say this: “Data in this study also support this result that T3SS may have some association with T6SS, but this association has yet to be studied in depth”
Author Response
Response to Reviewer 2
Dear reviewer:
We would like to thank you for your efforts in reviewing our manuscript, and providing many helpful comments and suggestions, which will all prove invaluable in the revision and improvement of our paper. We have studied your comments point by point, revised the manuscript accordingly. The detailed corrections are listed below. All the numbers of page and line refer to the revised manuscript file.
Authors presented research to investigate the role of Hcp of the T6SS in Acidovorax citrulli group 2 strains. The study was largely inspired by work done in Pseudomonas aeruginosa by Silverman et al. (2013). The hypothesis for the study is sound and should improve knowledge of T6SS in general and in A. citrulli. However, authors need to clarify certain experiments and explain their choice. I also think authors need to narrow their claims and only explain what they see from the experiment. A major concern in some earlier conclusions arose from the fact that authors did not first carry out in planta assays to understand strain growth patterns and what days after inoculation may approximate to populations of OD600= 1.5 at which T6SS is activated for WT and mutant strains. The log CFU values provided at 120 h and 18 d in Figure 1 for example (which were inoculated at OD600=0.3), could be shown alongside the approximate log CFU values of bacterial population at OD600= 1.5 to see if the population are comparable, or perhaps these effects were not due to T6SS at all if the bacterial population did not reach OD600=1.5 in planta within the days compared. Also, no graph or table for transcriptomics comparison was provided. I also cannot find Table S3 as stated by the authors. The discussion was difficult to follow, and authors need to refer to relevant sections in the paper while discussing. Because of the importance of this work, I am going to suggest acceptance with minor revision, hoping that the authors can address these points as soon as possible. Perhaps authors have better explanation that they wish to provide. I made relevant suggestions and I look forward to seeing an improved version and author’s response. Below are additional comments:
Response: Thank you for your constructive comments. In accordance with your suggestions, we have made revisions to our manuscript (lines 263-268).
“When the Optical Density value when measured at 600 nm wavelength in KB medium was 0.3 (OD600 = 0.3), the concentration of A. citrulli is approximately 3×108 CFU·mL-1 and log CFU·mL-1 is approximately 8.48. When the OD600 value is 1.5, the concentration is approximately 2×109 CFU·mL-1 and log CFU·mL-1 is approximately 9.30. As the result in Figure 2b and d, the bacterial population in watermelon cotyledons could not reach to 2×109 CFU·mL-1, the same population as OD600 = 1.5 in KB medium, until the cotyledons showed complete necrosis.”
This may due to the limited nutrition in cotyledons and the density control ability of A. citrulli. To study the activation of Hcp in limited nutrition, we used M9 medium in both qPCR and western blot assays. We found that even when the density reaches saturation (which also could not reach the same level as OD600 = 1.5) in M9 medium, Hcp was not activated. In cotyledons, Hcp was activated at 72 h although the bacterial population did not reach the same level as OD600 = 1.5. Based on these results, we think that host condition can activate T6SS, though the bacterial population did not reach to the same level as OD600 = 1.5, and led to the virulence and colonization of A. ctrulli Aac5.
We are sorry that we used hyperlinks by mistake to uploaded Table S3 causing it could not display. We have already re-uploaded the Table S3.
Line 15: Include a period (.) between “Acidovorax citrulli” and “Research”
Response: Thank you very much for your advice. It has been corrected in revised manuscript (line 15).
Line 21: The full meaning of hcp was earlier given but not tssm. Authors can provide the meaning in brackets or writing it out in full.
Response: Thank you very much for your advice. It has been revised as “The biofilm formation ability of ∆hcp, ∆tssm and ∆hcp∆tssm was lower than Aac5 when competition with prey bacteria but higher in KB and M9-Fe3+ medium. Deletion of hcp reduced the competition and survival ability of Aac5. Based on the results of western blotting and qRT-PCR analyses, Hcp is activated by cell density, competition, ferric iron, and the host plant. The expression levels of genes related to bacterial secretion systems, protein export, and several other pathways were significantly changed in the ∆hcp mutant compared to Aac5 when T6SS was activated at high cell density. Based on the transcriptome data, we found a few candidate effectors need further identification. The phenotypes, activated conditions and transcriptome data all supported the conclusion that: although there is only one T6SS gene cluster present in the A. citrulli group II strain Aac5, it related to multiple biological processes including colonization, growth ability, competition and biofilm formation.” (lines 26-36).
Line 48: change “amount of T6SS” to “number of T6SS”
Response: Thank you very much for your advice. It has been corrected in revised manuscript (line 66).
Line 50: change “and also studied” to “and is studied”
Response: Thank you very much for your advice. It has been corrected in revised manuscript (line 69).
Line 53: The statement “There are six sets of T6SS gene cluster in B. thailandensis” should come before the immediate previous sentence because the previous sentence compares T6SS in citrulli with B. thailandensis.
Response: Thank you very much for your advice. It has been corrected in revised manuscript (line 69).
Line 76-77: need to be reworded for flow. Throwing TssM in the beginning of the sentence brings an immediate break that is not in tandem with the previous sentence. Suggestion: “Additionally, since TssM is a key component of the T6SS apparatus (16) and present in the middle of A. citruli T6SS gene cluster (20), we used a tssm mutant as a T6SS deficient control.”
Response: Thank you very much for your advice. It has been revised in the manuscript (lines 134-136).
Line 85: “In a preliminary study”
Response: Thank you very much for your advice. It has been corrected in revised manuscript (line 91).
Line 71 -98: I suggest reorganization of this section thus:
“Hcp (Hemolysin co-regulated protein, also known as TssD) is both a special effector and a structural member of the inner tube of the T6SS [25]. Because of the secretory function and the assembly of the secretory channel with VgrG (Valine-glycine repeat 70 protein G; TssI) and PAAR (Pro-Ala-Ala-Arg-repeat-containing protein) [26, 27, 28], Hcp and VgrG can be used to study T6SS activation [29, 30, 31, 32, 33]. In a preliminary study, we found that the expression of one of the T6SS vgrG genes Aave_2840 was up-regulated when T3SS gene hrpE was absent in the A. citrulli group II strain Aac5 (unpublished transcriptome data). When Type 4 pilus (T4P) gene pilA was deleted, the expression of both hcp and vgrG was affected (unpublished transcriptome data). These unexpected data of T6SS gene indicate that T6SS of A. citrulli may have multiple biological functions, which needs to be proven.
At present, there is no report of the T6SS activated condition in A. citrulli, which is needed for the further research such as effector identification and transcription regulation of T6SS. In addition, due to the substantial differences between the two groups of A. citrulli strains and the horizontal transfer of T6SS genes [34, 35], the function of the T6SS in A. citrulli may vary by groups and strains. T6SS may also be related to other biochemical processed to realize its multiple functions.
In this study, we investigated the role of the key element hcp in virulence, competition, biofilm formation, growth ability and ferric ion absorption to better understand T6SS in A. citrulli. Because there are 12 VgrG but one Hcp in A. citrulli, we select Hcp as a marker for studying T6SS activation and secretion in this study. The transcriptional orientation of the hcp gene differs from that of all the other genes in the T6SS cluster of A. citrulli, which makes the hcp gene unique. Additionally, since TssM is a key component of the T6SS apparatus (16) and present in the middle of A. citruli T6SS gene cluster (20), we used a tssm mutant as a T6SS deficient control. The activation of hcp at different conditions was 93 determined by qPCR and western blotting to determine the activation condition of T6SS 94 in A. citrulli strain Aac5. In addition, the transcriptome data under the T6SS-activated condition was analyzed to study the possible relation of T6SS and other system and to find candidate effectors. These results will support further research on potential T6SS 97 effectors and the interaction between Hcp and VgrG proteins in A. citrulli. "
Response: Thank you for your constructive comments. It has been revised in the manuscript (lines 87-144).
"Line 103: Capitalize “We”
Response: Thank you very much for your advice. It has been corrected in revised manuscript (line 163).
Comment on Figure S1: Figure S1 is not labelled in the supplementary file. However, I assume the alignments before Figure S2 is Figure S1. It would be clearer if you place some pointers to the places of single mutations for each of the genes. It seems this is shown in fha but not others.
Response: Thank you very much for your advice. It has been revised in the supplementary files.
Line 107-109: Please be consistent. The locus tag of A. citrulli group II Aac5 was not mentioned while the locus tags for AAC00-1 and M6 were mentioned.
Response: Thank you very much for your advice. Sorry we are unable to provide the locus tag of A. citrulli group II strain Aac5. Because the genome of Aac5 has not been published yet, and there is no information of locus tag right now. We usually design primers based on the genome of AAC00-1, and use our Aac5 strain as template to amplify and sequence the fragments.
Comment on Figure 1: The figures appear blurred and could be improved.
Response: Thank you very much for your advice. It has been revised in the manuscript .
Figure 1a: Some of the words do not appear in full: Citrulli appears as citrull in some instances. Same for avenae.
Response: Thank you very much for your advice. It has been corrected in revised manuscript.
Figure 1b: This tree does not seem to be rooted with an outgroup, so it is difficult to say “A. citrulli and Hcp of T6SS-2, 4, and 6 of B. thailandensis were clustered in the same clade” as stated in line 113. This is especially because the bootstrap support for where the Hcp of citrulli strains diverged from the B. thailandensis is either 82 or 97, which suggests a strong support for divergence, not clustering. Please, root the tree with any of the species and see what you get.
Response: Thank you for your constructive comments. We rooted the tree with the outgroup Hcp2 of RIMD2210633 BAC62370.1, which is relatively low homology with other Hcps. As the figure above, we got similar results as before. The results (lines 187-189) and figure have been revised in manuscript.
Line 144: From Figure 2b, this statement is confusing:
“In the first 24 h, Δtssm, ΔhcpΔtssm, Δhcp, and the Hcp-complemented strain Δhcpcomp barely grew in the host tissue, while only the WT strain Aac5 maintained the initial cell concentration”
Figure 2b shows that the populations of Δtssm, ΔhcpΔtssm, Δhcp, and the Hcp-complemented strain Δhcpcomp in planta reduced, compared to the WT which maintained its initial inoculated population. So saying “barely grew in the host tissue” could be misleading.
Comment regarding Figure 2b: it is interesting that at 96 h and 120 h, Δtssm has higher population than all others.
Response: Thank you very much for your advice. It has been revised as “In the first 24 h, Δtssm, ΔhcpΔtssm, Δhcp, and the Hcp-complemented strain Δhcpcomp could not colonize the host tissue as the injected cell concentration, while only the WT strain Aac5 maintained the initial cell concentration (Figure 2b).” (lines 233-236).
Line 159-162: The conclusions from the virulence experiments, which should be correctly differentiated into in planta population assay and virulence assay, need to be improved. The authors concluded from their experiment that strain Aac overcame the loss of virulence by increasing the cell numbers when either hcp or tssm was deleted, and that the mechanism was not effective when both hcp and tssm were deleted. This statement is too general in my opinion.
Response: Thank you very much for your advice. We have already revised the statement in the manuscript (lines 252-279).
First, I expected the authors to delineate between in planta population growth assays and virulence, as they do not always mean the same (Please see Tardy et al. 2019: https://www.pnas.org/doi/pdf/10.1073/pnas.1901556116). The authors should also be familiar with this as they have demonstrated in their previous paper where they carried out different experiments for virulence assays and in planta population growth (See Guan et al. 2020; and Wang et al. 2022: https://www.frontiersin.org/articles/10.3389/fmicb.2021.783862/full). However, this is not clearly differentiated in this paper.
Secondly, from the results presented, the statement does not appear to be true. Figure 2a does not show any differences in virulence in my opinion. In Figure 2b, there was only reduced bacterial population in planta within 24 h when the initial inoculum was 104 CFU/ml. Subsequently, after 24 h, all strains increased their populations at different rates. The in planta bacterial population at 106 CFU/ml at 24 h is comparable between WT and Δtssm strains, while all others also increased as discussed by the authors (Figure 2d). In figure 2c, the pattern of change in virulence does not suggest that there was loss of virulence in the first place. Necrotic symptoms only appeared on the WT at 96 hpi, whereas it appears that necrosis development already began earlier on some of the other mutant strains at 72 hpi. Regardless, there was necrosis on all (except the CK) at 96h. At 120 h, it is difficult to differentiate between the level of necrosis.
I suggest using the virulence assay rating scale as in Guan et al. 2020 to approximate the ratings as they appear in figures 2a and c to improve this discussion.
Response: Thank you for your constructive comments. Actually, we conducted the virulence on cotyledons according to the standard by Bahar et al. (2009) and Hopkins and Thompson (2002) [36, 37] and in planta population growth assays at the same time.
Because there was no significant difference among the disease severity of the cotyledons injected by different strains, we did not display the results of virulence. We provide the results here and can add it into the revised manuscript later if necessary.
When the disease severity of the cotyledons injected by different strains showed no significant difference, the bacterial population showed certain differences. We think this may indicate that the virulence of per unit bacteria population has reduced. Based on these results, we suggested a hypothesis that Aac5 can overcome the minor loss of virulence by increasing the cell numbers when either hcp or tssm was deleted, but this mechanism was not effective when both hcp and tssm were deleted. Although the disease severity was not significantly different, the disease severity of ∆hcp∆tssm was still the lowest among all strains.
We agree that it is not prudent to draw such a conclusion, so we decided to remove this statement as the reviewer comment.
Figure 1 The disease severity standard by Bahar et al. (2009) and Hopkins and Thompson (2002) [36, 37]
[36] Hopkins, D. L., Thompson, C. M. Evaluation of Citrullus sp. germ plasm for resistance to Acidovorax avenae subsp. citrulli. Plant Dis. 2002, 86(1):61–64. https://doi.org/10.1094/PDIS.2002.86.1.61
[37] Bahar, O., Kritzman, G., and Burdman, S. Bacterial fruit blotch of melon: screens for disease tolerance and role of seed transmission in pathogenicity. Eur. J. Plant Pathol. 2009,123:71–83. https://doi.org/10.1007/s10658-008-9345-7
Figure 2 The disease severity of cotyledons caused by inoculation with WT, ∆hcp, ∆hcpcomp, ∆tssm, ∆hcp∆tssm, and sterile water (CK) at 24, 48, 72, 96, and 120 hour post inoculation (hpi). p=0.05. The strains were cultured until the logarithmic phase, resuspended in sterile water at 1 × 104 CFU·mL-1(a) and 1 × 106 CFU·mL-1 (b), and injected into the watermelon cotyledons.
Line 161-163: Concerning the statement of the authors: “Another possible explanation could be that when hcp or tssm were deleted separately, Aac5 lost the ability to control cell density.” This statement is too wild and does not appear to follow from the result presented. While there appears to be continued increase in bacteria population for strains Δtssm and Δhcp at 120 hpi for the initial inoculation concentration of 104 CFU/ml, this is not case for the 106 CFU/ml inoculation. In Figure 2d, at 120 hpi, the in planta population for strains Δtssm and Δhcp are lower than at 72 and 96 hpi. So, the results do not necessarily show loss of ability to control cell density.
Response: Thank you very much for your advice. We have already revised the statement as “When hcp or tssm were deleted separately, the decline phase delayed” (lines 251-252).
In the seed-to-seedling transmission assay, it would be important to take the bacterial population within 24 h of transferring to the 15 ml centrifuge tube as was done in the in planta population growth assay. I understand that the authors stated that the experiment was repeated three times, with 6 replicates per treatment, which should generally be robust. However, by not sampling bacterial population within 24 h, the authors cannot establish the initial population on the seeds, and it affects the inference that can be derived from this important experiment. From the previous section (figure 2d), although it was on cotyledons, the evidence suggests that at higher cfu/ml, there was comparable bacterial population for WT and Δtssm strain; similarly for Δhcp and Δhcpcomp at 24 h. Given that these strains were generated in this study, it makes sense to provide this information. Readers would not be sure what to expect on seeds, with respect to the initial populations of these strains on seeds. It is therefore difficult to infer anything from the drastic changes in bacterial populations within 3 days of each other as presented in Figure 2f.
Perhaps authors already have this result or have convincing explanation. Otherwise, I suggest repeating this experiment and replicated at least three times, and sampling bacterial populations within 24 h and 6dpi, including the other times used by authors.
Response: Thank you very much for your advice. We would like to make some explanations for the choice of this experiment. The aim of this experiment is to compare the ability of seed to seedling transmission and the population growth in whole seedlings after germination of all A. citrulli strains in this study.
We used all strains of the same concentration (108 CFU·mL-1) and volume (20 mL per 6 seeds) to soak the watermelon seeds (sterilized on the surface before soaking) in 50 mL centrifuge tubes at 220 rpm/min for 4 h. After thoroughly soaking, the seeds were fully dried by airing without washing and then transferred to 15 ml centrifuge tubes (6 seeds per tube, and 15 tubes per strain, 3 tubes were sampling for one strain each time) containing sterile absorbent cotton, filter paper, and sterile water. We considered that the same process was applied to the seeds, the same quantity of bacteria on the seed surface of seed should be obtained. This is one of the reasons that we did not sampling bacterial population within 24 h.
Additionally, at natural conditions, A. citrulli usually exist on the surface of seeds. When the seeds start germinating, the process of transmission from seed coats to seedlings also start at the same time. The time of sampling for seed to seedling transmission depended on the time of germination. Another reason we started sampling at 12 d but not 24 h or 6 d was that not all the seeds could germinate before 12 d (as a matter of fact, only CK could germinate before 6 d). The delay of seed germinations was especially obvious when inoculated with wild type Aac5 and ∆hcpcomp. So when the seeds inoculated with wild type Aac5 and ∆hcpcomp did not all germinate, they could not be comparable with the seedlings already germinated. As the seed coats were removed before the whole seedlings were sterilized on the surface, grinded and dissolve in ddH2O, the bacterial population was too few to determine before germination.
Due to the above reasons, we started sampling at 12 d but not 24 h or 6 d. Besides, the differences in size of seedlings between CK and treatments became were so big after 18 d that we could not continue sampling. Especially the seedlings treated by wild type Aac5, they stopped growing after 18 d.
As for the drastic changes within 3 days (the bacterial population increased about 104 in 3 days), the growth of population was consistent with the result of watermelon cotyledons. The growth of Δhcp, Δhcpcomp, Δtssm, and ΔhcpΔtssm was later than Aac5 in seed to seedling transmission, it was also consistent with the result of watermelon cotyledons.
Line 188-189: I think it is interesting to state that WT and Δhcpcomp behaved differently at OD 1.5 and 0.3
Response: Thank you very much for your advice. We have already revised this part to make this point (lines 310-314).
Line 192: “When co-cultured with DH5α, all strains formed more biofilm compared to without DH5α”: I do not see the well with DH5α alone. Or what was the control as suggested in line 551? Also check line 550: is it supposed to be ddH2O:DH5α = 30:1 or A. citrulli: ddH2O = 30:1?
Response: Thank you very much for your advice. It is our mistake. The control is supposed to be A. citrulli: ddH2O = 30:1.
In our pre-test of competition assay, when co-cultures at the ratio of A. citrulli: DH5α = 30:1 for 3 h, no DH5α could survive or form detectable biofilm. We chose this ratio in biofilm formation to make sure there was no biofilm formed by DH5α. This was one reason why we did not use and display the well with DH5α alone as control. The other reason was that even if we did have the wells with DH5α alone in this experiment at first, they could not be the control since the DH5α in the wells remained alive and could form biofilm. It has already been corrected in lines 736.
Growth ability assay: This section should have come much earlier in the experiment and as Figure 2. This could have helped to buttress the statement made on line 161-163 regarding cell density regulation, (however I still think my earlier statement remain valid for that section, especially when Δtssm is still showing different result).
Response: Thank you very much for your advice. We agree with the comments by reviewer. We revised the order of sections and the statement.
Line 224-
Figure 5: Please clearly state what “Ac” means in the description of the figure for b, d, f, h. Is it Ac or Aac5? For the other figures (a, c, e, g), authors used strain name. it is also difficult to differentiate what the bar colors stand for, since a single color is used to represent multiple phenotypes. I understand the difficultly because of multiple comparison. However, authors have not used the full range of colors available. Some of the colors used in Figure 3 (and more) could be used to differentiate between the bars.
Response: Thank you very much for your advice. Ac is short for A. citrulli. We revised it as “Ac strains”. The colors of bars have already been changed (line 378).
Figure 6a legend on the main graph: Aac5:pslb65, not A5. Same for Figure 6b legend on the main graph: the second Aac5 is written as A5. This figure has more colors representing different bars. However, two colors are used for two phenotypes. Authors need to state/explain using similar color for strains:pslb65 and their corresponding co-culture with ddH2O in Figure 6b.
Response: Thank you very much for your advice. It has already been corrected. The colors of bars have already been changed.
Line 240-244, Comment on Figure 6a: At 0 h post co-culture (pcc), WT ddH2O:pslb65 (a) already has significant difference in colony count than Aac5:pslb65 (c). It is therefore interesting that after 3h, they have comparable colony counts. This appears to be the same scenario with Δhcp and Δhcpcomp. It might be informative to include comparison of each treatment at 0h and 3h, with respect to the control ddH2O:pslb65. By doing this, the differences in co-culture between the WT Aac5 and the various mutants would become clearer.
Response: Thank you very much for your advice. It has already been revised in the text (lines 363-366) but not in the figure since the treatments were too much to add any other comparisons in the figure.
Line 288: This sentence is hanging: “To explore Hcp activation conditions, the following factors were examined.” What factors?
Response: Thank you very much for your advice. It has already been corrected (line 411).
Implications of the qRT-PCR result on earlier conclusions
The results from qRT-PCR put some of the earlier conclusions to question. From the qRT-PCR experiments, we see that T6SS was not activated at OD600 of 0.8. Meanwhile, experiments in 4.4 (in planta population assay, virulence and seed-to-seedling transmission) were done at OD600= 0.3. This puts further dent on the conclusions made in lines 159-163 since it is not clear that the T6SS was activated within the first 24 h of inoculation. Is it wise to compare at 24 hpi? It also makes the discussion on seed-to-seedling transmission difficult to agree with, especially since we don’t know how the variation in populations began in the early days and perhaps, T6SS may have been differentially activated in bacterial populations. This is same with growth ability assay and the previous in vitro assays.
Response: Thank you very much for your advice. As we can see in the results of growth ability, the strains have already reached the stationary phase when OD600 = 1.5. The bacteria of this phase had terrible growth ability even begin to decline. Based on this, we could not use the bacteria of OD600 = 1.5 to conduct any experiment that need the growth of bacteria. About in planta population and virulence assay, we demonstrated that hcp could also be activated by 48 h and 72 h in planta when the initial concentration was 106 CFU·mL-1 (OD600 = 0.3 equates to 3×108 CFU·mL-1 and OD600 = 1.5 equates to 2×109 CFU·mL-1). And when hcp activated at 72 h in planta, the concentration approached 3×108 CFU·mL-1 but was still significantly lower than 2×109 CFU·mL-1. So we considered that hcp in planta at the concentration about 108 CFU·mL-1. Thus the comparison we did in population and virulence assay could still make sense. Since the population of strains did not all reach 108 CFU·mL-1 until 18 d in seed to seedling transmission of figure 2f, to compare the population at 18 d could explain the function of hcp and T6SS.
As for the differences among strains in phenotypes when hcp was not activated, they could be used to compared to the differences under activated condition for the understanding of hcp. We guess that the differences when hcp was not activated may because some other systems were affected when deleted hcp even under inactivated condition.
The qPCR experiments also focused on the strains grown under KB medium (without antibiotic resistance) (4.8). In 4.1, the authors stated that media used for experiments had antibiotic resistance. Given that many of these experiments were carried out at OD=0.3, it would be important to see the OD at which T6SS is activated under the various antibiotic resistance conditions where these experiments were carried out. Antibiotics have effects on many aspects of bacterial growth, including biofilm formation (https://www.frontiersin.org/articles/10.3389/fmicb.2020.02109/full), secretion systems (https://www.sciencedirect.com/science/article/pii/S0167488914000512), etc. Perhaps, this will provide answers to some of the differences (or no differences) seen in the populations in the different experiments. Perhaps, T6SS was not activated at 0 h at OD OD=0.3 in Figures 2-5.
Response: Thank you very much for your advice. As the comment of reviewer, we have considered this. We only used antibiotics in competition assay of counting colonies in plates and bacteria culture before experiments. During the experiments such as growth ability, biofilm formation, virulence and colonization, we did not use any antibiotics. We should express this clearly so we revised it in the manuscript (lines 652-654). Besides, we also conducted qPCR in KB with Amp of OD600 = 0.3 and 1.5, the results were similar.
Comment on Section 2.8: The authors need to include a supplementary figure showing comparable variations in gene regulations under the various conditions. Figure 9 as presented shows nothing in that respect. Authors stated that vgrG in Δhcp was downregulated, but I could not find the results where these were compared. I also cannot find Table S3 in the files available to me. This applies to most conclusions in this section.
Response: Thank you very much for your advice. We are sorry that we used hyperlinks by mistake to uploaded Table S3 causing it could not display. We have already reuploaded Table S3 correctly.
Line 351, Figure 9 legend: remove the first (c ) and (d)
Response: Thank you very much for your advice. It has already been corrected.
Comments on discussion:
This section needs to be thoroughly organized so that paragraphs address major points at a time. The discussion on many results are packed together in a single paragraph which makes it difficult to follow. Also, authors should reference the particular figures addressed in the discussion. I believe some of the discussion can also be improved after the earlier comments are addressed.
Response: Thank you very much for offering valuable advice. We have carefully revised the discussion (lines 490-637).
Line 360: Prefix TssM with “Deletion of tssm, the gene responsible for…”
Response: Thank you very much for your advice. It has already been corrected (line 499).
Line 576-577: No mention of plant before throwing this sentence: “The cotyledons were collected at 24, 48, and 72 h after injection at high concentration (106 CFU·ml−1) for RNA extraction.”. Add: “For in planta studies” or variation of author’s choice.
Response: Thank you very much for your advice. It has already been corrected (line 762).
Line 398-399: Authors stated “Therefore, Hcp and the T6SS might not be activated under this growth condition.”. So why did you use lower ODs for Figure 2 etc if you knew this? The authors further stated:“It is critical to define the conditions necessary for T6SS activation for further research, such as screening, identifying, and characterizing T6SS effectors”. I agree.
Response: Thank you very much for your advice. The low OD value was used to compare with high OD to see the differences of activated and inactivated condition. And we want to see if the expression of Hcp increased gradually as the density increased (OD600=0.3, 0.8, 1.0 and 1.5) or activated by specific cell density.
Line 406: Which figure: “The cell density at 72 hpi in the host had not reached the same level as that of OD600=1.5 in KB medium…”
Response: Thank you very much for your advice. It has already been corrected (line 564).
Line 412-413: I am not sure you can broadly say these: “Based on the above results, T6SS and its key element Hcp in A. citrulli are involved in watermelon cotyledon colonization and seed-to-seedling transmission, interspecies and intraspecies competition, and regulation of cell density.” Please see or address earlier comments on Figure 2.
Response: Thank you very much for your advice. It has already been revised (lines 575-577).
Line 415: Include figure: “but in Aac5 we found that the production and secretion of Hcp was promoted by ferric iron (Figure?).
Response: Thank you very much for your advice. It has already been corrected (line 579).
Line 422-442: Authors need to provide a table or graph comparing transcriptomics results.
Response: We are sorry that we used hyperlinks by mistake to uploaded Table S3 causing it could not display. We have already reuploaded Table S3 correctly.
Line 428-430: I can’t see this comparison: “Among the differentially expressed genes under the cell density activated condition, we found twelve genes encoding secreted proteins predicted by SecretomeP-2.0.”
Response: We are sorry that we used hyperlinks by mistake to uploaded Table S3 causing it could not display. We have already reuploaded Table S3 correctly.
Line 432-463: Authors need to stop starting sentences with conjunctions such “But”, “And”:
“But further experiments such as secretion assay and function analyses need to be conducted to identify it as a T6SS effector. And there were five genes encoding secreted proteins that showed certain homology to effectors proteins recorded in SecReT6. But the homogeneity of some genes was not high enough, they also need further experiments of identification.”
Response: Thank you very much for your advice. It has already been corrected (lines 616-617).
Line 425-426: where can I see this data and why did authors say this: “Data in this study also support this result that T3SS may have some association with T6SS, but this association has yet to be studied in depth”
Response: Thank you very much for your advice. It has already been corrected (lines 605-610).
If there are any other modifications we could make, we would like very much to modify them and we really appreciate your help. Thank you very much for your help.
Sincerely yours,
Yuwen Yang, Tingchang Zhao and Nuoya Fei
State Key Laboratory for Biology of Plant Diseases and Insect Pests, Institute of Plant Protection, Chinese Academy of Agricultural Sciences
China, Beijing, Yuanmingyuan, 2
100193 Beijing
China
Email: yangyuwen@126.com, zhaotgcg@163.com, fei_nuoya@126.com

Reviewer 3 Report
The manuscript describes the study of the role of T6SS in Acidovorax citrulli in various aspects including pathogenesis, biofilm formation, competition and ferric ion absorption. The authors provide a good introduction of T6SS and present large amount of data in the manuscript. However, the results seems to be difficult to suggest the role of hcp/tssm in those physiological responses or simply due to the compromised overall fitness due to the loss of these genes. In the end, it is hard to be persuaded that the results support the claims from the authors.
Major comments:
1. Hcp and Tssm are the core proteins in T6SS, it is expected the phenotypes should be similar (loss of these genes are equivalent to the loss of whole T6SS functionality, as A. ctrulli only has one copy of hcp gene). However most of the results between these mutants are contradicted. The design of experiments is also unusual, for example, the tssm mutant is not complemented (e.g. tssmcomp), but rather a double hcptssm mutant is used. This makes readers hard to draw a conclusion from the results presented.
2. The strain hcpcomp is not able to complemented the hcp phenotypes in the hcp mutant (For example, Fig 1f, 2a, 2c), which makes it difficult to suggest the role of hcp.
3. Although there are showing statistically different in the graphs presented, there are very minor differences when considering the error bars. Examples including Fig. 2d, 3a, 5, 6. It is hard to be persuaded there are real difference in the results that the authors can claim and make conclusion.
4. The qPCR design in Fig 7 is uncommon. The comparison should be done by only using WT instead of hcpcomp strain. The complemented plasmid may have different copy number which is unfair to be compared with WT Aac5.
Minor comments:
1. Section 2.2 does not present any figures in the main text or supplemented information to show the results of mutant confirmation. "hcpcomp" is not a common presentation and it is not explained (it is believed to be the complemented strain of hcp)
2. Line 161-162 may not be a well-supported statement, since the same strain will have the same immunity protein which may be irrelevant for cell density control through T6SS.
Author Response
Response to Reviewer3
Dear reviewer:
We feel great thanks for your professional review work on our article. According to your comments, we have revised the manuscript extensively and make some additional explanations to the comments point by point. If there are any other modifications we could make, we would like very much to modify them and we really appreciate your help.
The manuscript describes the study of the role of T6SS in Acidovorax citrulli in various aspects including pathogenesis, biofilm formation, competition and ferric ion absorption. The authors provide a good introduction of T6SS and present large amount of data in the manuscript. However, the results seems to be difficult to suggest the role of hcp/tssm in those physiological responses or simply due to the compromised overall fitness due to the loss of these genes. In the end, it is hard to be persuaded that the results support the claims from the authors.
Major comments:
- Hcp and Tssm are the core proteins in T6SS, it is expected the phenotypes should be similar (loss of these genes are equivalent to the loss of whole T6SS functionality, as A. ctrulli only has one copy of hcp gene). However most of the results between these mutants are contradicted. The design of experiments is also unusual, for example, the tssm mutant is not complemented (e.g. tssmcomp), but rather a double hcptssm mutant is used. This makes readers hard to draw a conclusion from the results presented.
Response:
Thank you for your comments. This is our mistake that we should explain more clearly in introduction and discussion. We have improved it in the revised.
Hcp is both a special effector and a baseplate complex (BC) structural member while TssM is inner membrane proteins of T6SS membrane complex (MC). The BC assembles independently of the MC. And the transcriptional orientation of the hcp gene differs from that of all the other genes in the T6SS cluster of A. citrulli, which makes the hcp gene unique. In our secretion assay and Pei’s article, the secretion of Hcp reduced but not fully lost in ∆tssm (Figure 8), which might also indicate that Hcp was an independent component in T6SS and could still work outside the cell in ∆tssm. In Tian’s article, the phenotypes of different T6SS structural gene deletion mutant showed differences (ΔvasD, ΔimpK, ΔimpJ and ΔimpF affected the seed-to-seedling transmission of melon while other 13 mutant strains of T6SS core genes not). The T6SS assembly and the mechanism of action are conserved across species, but the repertoire of secreted toxic effectors and cognate immunities and their regulatory mechanisms vary by strains [44,45].
Based on these, we came up with the hypothesis that this difference may due to the activated condition and the difference of transcription and designed a series of experiments. In this study, we mainly used tssm as a control strain since the phenotypes of ∆tssm has already reported in both Tian et al. (phenotypes of pathogenicity) and Pei et al. (phenotypes of competition), so we did not construct the complementary strain of tssm. This was also supported by transcriptome analysis that only one T6SS core component VgrG was affected by deletion of hcp. These could explain the differences of hcp and tssm phenotypes. The phenotypes of ∆hcp and ∆tssm were basically consistent in most activated conditions (cell density of OD600 = 1.5 in KB medium and host condition after 72 h) we demonstrated (only except the phenotypes in ferric ion). As for the double mutant ∆hcp∆tssm, we think it can represent the phenotypes of losing T6SS better than ∆hcp and ∆tssm if hcp and tssm behave in different way. Now that we have obtained the differential phenotypes in different activated conditions and different strains, it proves that our hypothesis may be correct.
We agree with the comments by reviewer and the manuscript has been modified (lines 497-501, 513-527).
- The strain hcpcomp is not able to complemented the hcp phenotypes in the hcp mutant (For example, Fig 1f, 2a, 2c), which makes it difficult to suggest the role of hcp.
Response: Thank you for your constructive comments. This is indeed a controversial issue that we should explain more clearly in results and discussion.
Based on qPCR and western blotting, we demonstrated that the expression of hcp in ∆hcpcomp is stable. The survival and competition ability of ∆hcpcomp is consistent with WT strain Aac5. whereas other phenotype such as biofilm formation, colonization, ferric iron absorption and growth ability of ∆hcpcomp partial recovery, which may be due to two reasons: (1) in some conditions, T6SS was not activated. This was supported by that the phenotypes of wild type strain and ∆hcpcomp basically consistent in most activated conditions (cell density of OD600 = 1.5 and host condition after 72 h) we demonstrated (only except the phenotypes in ferric ion). The other reason may be that (2) Hcp in complementary strain was in an expression vector which was demonstrated that could express without activated even not in the genome. Although the expression plasmid had a low copy number, as our pre-test, when hcp was activated in wild type strain the expression could still not reach the level of ∆hcpcomp (except co-cultured with DH5α, only under this condition, expression of hcp in Aac5 was significantly higher than that in ∆hcpcomp). But the differential expression of hcp between wild type and ∆hcpcomp did reduced significantly in activated condition (Figure S3b-h). The difference expression of hcp should responsible for the phenotypes.
We agree with the comments by reviewer and the manuscript has been modified (lines 585-601).
- Although there are showing statistically different in the graphs presented, there are very minor differences when considering the error bars. Examples including Fig. 2d, 3a, 5, 6. It is hard to be persuaded there are real difference in the results that the authors can claim and make conclusion.
Response: Thank you very much for your advice. We would like make some explanations to the results and differences. The y-axis of Figure 2, 5, and 6 was calculated by log10CFU·g-1 or log10CFU·mL-1because the digits of original values was too many, so the error bars between bars might seem small and differences seemed less than those of original values (CFU·g-1 or CFU·mL-1).
For accuracy, the significance was calculated by both GraphPad and SPSS. In the figures of manuscript, the significance by SPSS (one way ANOVA, Duncan’s, P=0.05 or 0.01) were displayed. In view of your doubts, we used T-test to compared every treatment strain with WT and got similar results. The original data can be provided if necessary.
- The qPCR design in Fig 7 is uncommon. The comparison should be done by only using WT instead of hcpcomp strain. The complemented plasmid may have different copy number which is unfair to be compared with WT Aac5.
Response: Thank you very much for your advice. We agree with that WT strain is better so the manuscript has been modified as only using WT (Figure 7).
And we would like make some additional explanations. At first, we designed the activated condition assay based on the expression difference between wild type and complementary strain. Thus, we regarded ∆hcpcomp as a positive control of activation in most qPCR and western blot assay. We think activated condition can be proved by both the difference expression of hcp in wild type of different conditions and difference expression of hcp between wild type and ∆hcpcomp of activated conditions (OD600=1.5, 72 hpi in host, and co-cultured with E. coli DH5α) less than that of inactivated conditions. Reduction of difference between wild type and ∆hcpcomp uploaded only as supplementary materials.
Minor comments:
- Section 2.2 does not present any figures in the main text or supplemented information to show the results of mutant confirmation. "hcpcomp" is not a common presentation and it is not explained (it is believed to be the complemented strain of hcp)
Response: Thank you very much for your advice. The results of mutant confirmation have been provided as supplementary materials (Supplementary Files, Figure S2b). ∆hcpcomp is short for ∆hcp complemented strain. The explain of complemented strain has been added (line 199).
- Line 161-162 may not be a well-supported statement, since the same strain will have the same immunity protein which may be irrelevant for cell density control through T6SS.
Response: Thank you very much for your advice. We have already revised the statement in the manuscript (lines 251-252).
If there are any other modifications we could make, we would like very much to modify them and we really appreciate your help. Thank you very much for your help.
Sincerely yours,
Yuwen Yang, Tingchang Zhao and Nuoya Fei
State Key Laboratory for Biology of Plant Diseases and Insect Pests, Institute of Plant Protection, Chinese Academy of Agricultural Sciences
China, Beijing, Yuanmingyuan, 2
100193 Beijing
China
Email: yangyuwen@126.com, zhaotgcg@163.com, fei_nuoya@126.com
Round 2
Reviewer 3 Report
No additional comments